# Generalized Linear Bandits:
# Almost Optimal Regret with One-Pass Update

**Yu-Jie Zhang[1], Sheng-An Xu[2,3], Peng Zhao[2,3], Masashi Sugiyama[1,4]**
[1] RIKEN AIP, Tokyo, Japan
[2] National Key Laboratory for Novel Software Technology, Nanjing University, China
[3] School of Artificial Intelligence, Nanjing University, China
[4] The University of Tokyo, Chiba, Japan

## Abstract

We study the generalized linear bandit (GLB) problem, a contextual multi-armed bandit framework that extends the classical linear model by incorporating a non-linear link function, thereby modeling a broad class of reward distributions such as Bernoulli and Poisson. While GLBs are widely applicable to real-world scenarios, their non-linear nature introduces significant challenges in achieving both computational and statistical efficiency. Existing methods typically trade off between two objectives, either incurring high per-round costs for optimal regret guarantees or compromising statistical efficiency to enable constant-time updates. In this paper, we propose a jointly efficient algorithm that attains a nearly optimal regret bound with $\mathcal{O}(1)$ time and space complexities per round. The core of our method is a tight confidence set for the online mirror descent (OMD) estimator, which is derived through a novel analysis that leverages the notion of mix loss from online prediction. The analysis shows that our OMD estimator, even with its one-pass updates, achieves statistical efficiency comparable to maximum likelihood estimation, thereby leading to a jointly efficient optimistic method.

## 1 Introduction

Stochastic multi-armed bandits [Robbins, 1952] represent a fundamental class of sequential decision-making problems where a learner interacts with environments by selecting actions (or arms) and receiving feedback in the form of rewards. In this paper, we study the contextual multi-armed bandit problem under the framework of generalized linear models (GLMs). In this setting, each action is characterized by a contextual feature vector $\mathbf{x} \in \mathcal{X}_t \subset \mathbb{R}^d$, where the arm set $\mathcal{X}_t$ may vary over time. More specifically, the learning process can be seen as a $T$ round game between the learner and environments: at each round $t$, the learner selects an action $X_t \in \mathcal{X}_t$ and then observes a stochastic reward $r_t \in \mathbb{R}$ generated according to a GLM (see Definition 2.1). The goal of the learner is to maximize the cumulative expected reward obtained over the time horizon $T$. Under the GLM model, the expectation of the reward satisfies $\mathbb{E}[r_t \mid X_t] = \mu(X_t^\top \theta_*)$, where $\mu : \mathbb{R} \to \mathbb{R}$ is a non-linear link determined by the GLM model and is known to the learner. The unknown part is the underlying parameter $\theta_* \in \mathbb{R}^d$, which needs to be estimated from the observed action-reward pairs.

Compared with the classical linear case [Abbasi-Yadkori et al., 2011], the generalized linear bandit (GLB) framework allows for a richer class of reward distributions, including Gaussian, Bernoulli, and Poisson distributions. This flexibility enables the modeling of various real-world tasks, such as recommendation systems [Li et al., 2010] and personalized medicine [Tewari and Murphy, 2017], where the feedback is binary (Bernoulli) or count-based (Poisson) and inherently non-linear. Besides

---

*Correspondence: Peng Zhao <zhaop@lamda.nju.edu.cn>

39th Conference on Neural Information Processing Systems (NeurIPS 2025).

**Table 1:** Comparison of regret guarantees and computational complexity per round for GLBs. Here, $\kappa_* = 1/\left(\frac{1}{T}\sum_{t=1}^{T}\mu'(\mathbf{x}_{t,*}^\top\theta_*)\right)$ is the slope at the optimal action $\mathbf{x}_{t,*} = \arg\max_{\mathbf{x}\in\mathcal{X}_t}\mu(\mathbf{x}^\top\theta_*)$, with $\kappa_* \leq \kappa$ (see Section 2 for details). † indicates the amortized time complexity, i.e., average per-round cost over $T$ rounds.

| Method | Regret | Time per Round | Memory | Jointly Efficient |
|---|---|---|---|---|
| GLM-UCB [Filippi et al., 2010] | $\mathcal{O}(\kappa(\log T)^{\frac{3}{2}}\sqrt{T})$ | $\mathcal{O}(t)$ | $\mathcal{O}(t)$ | ✗ |
| GLOC [Jun et al., 2017] | $\mathcal{O}(\kappa\log T\sqrt{T})$ | $\mathcal{O}(1)$ | $\mathcal{O}(1)$ | ✗ |
| OFUGLB [Lee et al., 2024, Liu et al., 2024] | $\mathcal{O}(\log T\sqrt{T/\kappa_*})$ | $\mathcal{O}(t)$ | $\mathcal{O}(t)$ | ✗ |
| RS-GLinCB [Sawarni et al., 2024] | $\mathcal{O}(\log T\sqrt{T/\kappa_*})$ | $\mathcal{O}((\log t)^2)^\dagger$ | $\mathcal{O}(t)$ | ✗ |
| GLB-OMD (Theorem 2 of this paper) | $\mathcal{O}(\log T\sqrt{T/\kappa_*})$ | $\mathcal{O}(1)$ | $\mathcal{O}(1)$ | ✓ |

its practical appeal, the study of GLB lays theoretical foundations for other sequential decision-making problems, such as function approximation in RL [Wang et al., 2021, Li et al., 2024], safe exploration [Wachi et al., 2021], and dynamic pricing [Chen et al., 2022, Xu and Wang, 2021].

The non-linearity of the link function raises significant concerns regarding both computational and statistical efficiency in GLBs. A canonical solution to GLB is the GLM-UCB algorithm [Filippi et al., 2010], which belongs to the family of UCB-type methods [Agrawal, 1995, Auer, 2002]. At each iteration $t \in [T]$, the algorithm estimates the true parameter $\theta_*$ using maximum likelihood estimation (MLE) based on the historical data $\{(\mathbf{x}_s, r_s)\}_{s=1}^{t-1}$ and yields an estimator $\theta_t$, which is further used for constructing the upper confidence bound for arm selection. As shown in Table 1, GLM-UCB achieves nearly optimal regret bound in terms of the dependence on $T$. However, its reliance on MLE incurs a computational burden: it requires storing all historical data with $\mathcal{O}(t)$ space complexity and solving an optimization problem with $\mathcal{O}(t)$ time complexity at each round $t$. Besides, in terms of the statistical efficiency, the non-linearity of the link function introduces a notorious constant $\kappa = 1/\inf_{\mathbf{x}\in\cup_{t=1}^{T}\mathcal{X}_t,\theta\in\Theta}\mu'(\mathbf{x}^\top\theta)$ into the regret bound of GLM-UCB (see Section 2 for details), where $\mu'$ denotes the derivative of $\mu$. In several applications of GLBs, such as logistic bandits $\left(\mu(z) = 1/(1+e^{-z})\right)$ and Poisson bandits $\left(\mu(z) = e^z\right)$, the $\kappa$ term can grow exponentially with the norm of the parameter $\|\theta_*\|_2$, severely affecting the theoretical performance.

Over the past decade, extensive efforts have been devoted to enhancing the computational or statistical efficiency of GLBs. However, as summarized in Table 1, how to develop a jointly efficient method that achieves both low computation cost and strong statistical guarantees remains unclear. For GLBs, Jun et al. [2017] developed computationally efficient algorithms with one-pass update, but their regret bound scales linearly with the potentially large constant $\kappa$. More recently, by leveraging the self-concordance of the loss, several works [Lee et al., 2024, Sawarni et al., 2024, Liu et al., 2024, Clerico et al., 2025] proposed statistically efficient methods that achieve improved dependence on $\kappa$; however, these approaches are still based on the MLE, which has high computation cost.

**Our Results.** This paper proposes a jointly efficient algorithm for GLBs that achieves an improved regret bound in terms of $\kappa$ with constant time and space complexities per round as shown in Table 1. This advance roots in the construction of a tight confidence set for the online mirror descent (OMD) estimator used for performing the UCB-based exploration. We show that the OMD estimator, even though updated in a one-pass fashion, can still match the statistical efficiency of the MLE by carefully addressing the non-linearity of the link function. Here, "one-pass" refers to processing each data point only once, without storing past data. We also note that OMD-based online estimators have been used to develop jointly efficient algorithms in the logistic bandit setting [Faury et al., 2022, Zhang and Sugiyama, 2023, Lee and Oh, 2024], a special case of GLBs with Bernoulli rewards. However, their analyses of the confidence set rely heavily on the specific structure of the logistic link function $\mu(z) = 1/(1+e^{-z})$, which limits their applicability to the more general GLB setting.

**Technical Contribution.** Our main technical contribution is a new analysis of the estimation error of the OMD estimator. The analysis is based on the concept of the *mix loss*, which has been used in full-information online learning to achieve fast-rate regret minimization [Vovk, 2001]. Here, we show that it provides a natural way to bridge the

**Table 2:** Jointly efficient methods for logistic bandits.

| Method | Regret | Time | Memory |
|---|---|---|---|
| (ada)-OFU-ECOLog [Faury et al., 2022] | $\mathcal{O}(\log T\sqrt{T/\kappa_*})$ | $\mathcal{O}(\log t)$ | $\mathcal{O}(1)$ |
| OFUL-MLogB [Zhang and Sugiyama, 2023] | $\mathcal{O}((\log T)^{3/2}\sqrt{T/\kappa_*})$ | $\mathcal{O}(1)$ | $\mathcal{O}(1)$ |
| GLB-OMD (Theorem 2) | $\mathcal{O}(\log T\sqrt{T/\kappa_*})$ | $\mathcal{O}(1)$ | $\mathcal{O}(1)$ |

gap between the OMD estimator and the true parameter $\theta_*$, thereby enabling the construction of tight confidence sets for bandit online learning. Our new analysis not only generalizes the OMD-based approach to the broader GLBs but also improves upon the state-of-the-art for logistic bandits. As shown in Table 2, the jointly efficient method [Faury et al., 2022] requires $\mathcal{O}(\log t)$ time per round and an adaptive warm-up strategy to achieve optimal regret. Zhang and Sugiyama [2023] reduces the time complexity to $\mathcal{O}(1)$ but incurs an extra $\mathcal{O}(\sqrt{\log T})$ factor in the regret. In contrast, our refined analysis yields a tighter error bound for the OMD estimator, allowing our method to achieve improved regret and low computation cost without warm-up rounds. Details are provided in Section 4.

We were made aware that mix-loss–based analyses have been independently developed in two very recent concurrent works for constructing tight confidence sets. Specifically, Kirschner et al. [2025] developed confidence sets based on the sequential likelihood ratios mixing technique, and Clerico et al. [2025] proposed several confidence sets for GLMs. While conceptually related, these works focus on the batch setting, where all historical data are repeatedly accessed, leading to substantial computational overhead. In contrast, our confidence set is based on the OMD estimator with a one-pass update. This difference leads to a distinct formulation of the mix loss and a tailored analysis to quantify its gap relative to the time-varying OMD estimator. Details are provided in Section 4.2.

## 2   Preliminary

This section provides background on the GLB problem, including its formulation, underlying assumptions, and closely related previous research. In the rest of the paper, for a positive semi-definite matrix $H \in \mathbb{R}^{d \times d}$ and vector $\mathbf{x} \in \mathbb{R}^d$, we define $\|\mathbf{x}\|_H = \sqrt{\mathbf{x}^\top H \mathbf{x}}$ and $\|\mathbf{x}\|_2$ as the Euclidean norm. For the function $f : \mathbb{R} \to \mathbb{R}$, its first and second derivatives are denoted by $f'$ and $f''$, respectively.

### 2.1   Problem Formulation and GLM-UCB [Filippi et al., 2010]

The GLB problem considers a $T$-round sequential interaction between a learner and the environment. At each round $t \in [T] \triangleq \{1, \ldots, T\}$, the learner selects an action $X_t \in \mathcal{X}_t \subset \mathbb{R}^d$ from the feasible domain and then receives a stochastic reward $r_t \in \mathbb{R}$. We use the notation $\mathcal{X}t$ to indicate that the arm set may vary over time, capturing many practical scenarios where available options change dynamically. For instance, in product recommendation systems, items can be added or removed, requiring the algorithm to adapt accordingly. Besides, we denote the learner's action by $X_t$ to emphasize its stochastic nature, which may depend on past data captured by the filtration $\mathcal{F}_t = \sigma(X_1, r_1, \ldots, X_{t-1}, r_{t-1})$. In GLBs, conditioned on $\mathcal{F}_t$, the reward $r_t$ follows a canonical exponential family distribution with the natural parameter given by the linear model $z_t = X_t^\top \theta_*$.

$$\Pr\left[r_t \mid z_t = X_t^\top \theta_*, \mathcal{F}_t\right] = \exp\left(\frac{r_t z_t - m(z_t)}{g(\tau)} + h(r_t, \tau)\right). \tag{1}$$

Here, $\theta_* \in \mathbb{R}^d$ is a $d$-dimensional vector unknown to the learner. The function $h(r, \tau)$ is the base measure, which provides the intrinsic weighting of the variable $r$, while $m(z)$ is twice continuously differentiable function used for normalizing the distribution. Besides, $g : \mathbb{R} \to \mathbb{R}$ is the dispersion function controlling the variability of the distribution and $\tau \in \mathbb{R}$ is a known parameter. The expectation and variance of the exponential family distribution can be calculated as $\mathbb{E}[r \mid z] = \mu(z) \triangleq m'(z)$ and $\mathrm{Var}(r \mid z) = g(\tau) m''(z)$ [Wainwright and Jordan, 2008, Proposition 3.1]. Common examples of (1) include Gaussian, Bernoulli, and Poisson distributions. The goal of the learner is to maximize the cumulative expected reward, which is equivalent to minimizing the regret,

$$\mathrm{REG}_T = \sum_{t=1}^{T} \mu(\mathbf{x}_{t,*}^\top \theta_*) - \sum_{t=1}^{T} \mu(X_t^\top \theta_*),$$

where $\mathbf{x}_{t,*} = \arg\min_{\mathbf{x} \in \mathcal{X}_t} \mu(\mathbf{x}^\top \theta_*)$ is the optimal action. Besides, we have the following standard boundness assumptions used in the GLB literature [Filippi et al., 2010].

**Assumption 1** (bounded domain). The set $\bigcup_{t \in [T]} \mathcal{X}_t$ is bounded such that $\|\mathbf{x}\|_2 \leq 1$ for all $\mathbf{x} \in \mathcal{X}_t$, $t \in [T]$ and the parameter $\theta_*$ satisfies $\|\theta_*\|_2 \leq S$ for some constant $S > 0$ known to the learner.

**Assumption 2** (bounded link function). The link function $\mu$ is twice differentiable over its feasible domain. Moreover, there exist constants $c_\mu > 0$ and $L_\mu > 0$ such that $c_\mu \leq \mu'(z) \leq L_\mu$ for all $z \in [-S, S]$. Consequently, the function $m$ is strictly convex and $\mu$ is strictly increasing.

**GLM-UCB Method and Potentially Large Constant.** The canonical algorithm for the GLB problem is GLM-UCB [Filippi et al., 2010], which resolves the exploration-exploitation trade-off with an upper-confidence-bound strategy [Agrawal, 1995]. Under Assumptions 1, 2 and an additional condition that the reward $r_t$ is non-negative and almost surely bounded for all $t \in [T]$, GLM-UCB achieves the regret of $\mathcal{O}\left(\kappa(\log T)^{\frac{3}{2}}\sqrt{T}\right)$, where the $\mathcal{O}(\cdot)$-notation is used to highlight the dependence on $\kappa$ and time horizon $T$. The dependence on the horizon $T$ matches that of the linear case, where the $\widetilde{\mathcal{O}}(\sqrt{T})$ rate is nearly optimal [Dani et al., 2008]. The bottleneck lies in its *linear* dependence on the constant $\kappa$, which is defined by

$$\kappa \triangleq \frac{1}{c_\mu} = \frac{1}{\inf_{\mathbf{x} \in \mathcal{X}_{[T]}, \theta \in \Theta} \mu'(\mathbf{x}^\top \theta)} \quad \text{and} \quad \kappa_* \triangleq \frac{1}{\frac{1}{T}\sum_{t=1}^{T} \mu'(\mathbf{x}_{t,*}^\top \theta_*)},$$

where $\Theta = \{\theta \in \mathbb{R}^d \mid \|\theta\|_2 \leq S\}$ and $\mathcal{X}_{[T]} = \bigcup_{t \in [T]} \mathcal{X}_t$. In the above we also define $\kappa_*$ to reflects the local curvature at the optimal actions. The linear dependence on $\kappa$ in GLM-UCB is generally undesirable, as $\kappa$ can be prohibitively large in practice. Notable examples include the Bernoulli distribution with $\mu(z) = 1/(1 + e^{-z})$ and the Poisson distribution with $\mu(z) = e^z$, for which $\kappa = \mathcal{O}(e^S)$, growing exponentially with the parameter-norm bound $S$.

## 2.2 New Progress with Self-Concordance

The undesirable linear dependence on $\kappa$ has motivated the development of algorithms with improved theoretical guarantees. By leveraging the self-concordance of the loss, rooted in convex optimization and later used in the analysis of logistic regression [Bach, 2010], recent studies [Russac et al., 2021, Lee et al., 2024, Sawarni et al., 2024] have derived regret bounds with substantially reduced dependence on $\kappa$ for GLB. Following this line, we also adopt the self-concordance assumption here.

**Assumption 3** (Self-Concordance). The link function satisfies $|\mu''(z)| \leq R \cdot \mu'(z)$ for all $z \in \mathbb{R}$.

Assumption 3 holds for many widely used GLMs. For GLMs where the reward is almost surely bounded in $[0, R]$, the link function satisfies Assumption 3 with coefficient $R$ [Sawarni et al., 2024]. For example, the Bernoulli distribution is $1$-self-concordant. Many unbounded GLMs also satisfy self-concordance, including the Gaussian distribution ($R = 0$), Poisson distribution ($R = 1$), and Exponential distribution ($R = 0$). Leveraging self-concordance, Lee et al. [2024] and Sawarni et al. [2024] established improved regret bounds of order $\widetilde{\mathcal{O}}(\sqrt{T/\kappa_*})$. In these results, the potentially large constant $\kappa_*$ appears at the denominator, which largely improves the $\widetilde{\mathcal{O}}(\kappa\sqrt{T})$ bound by Filippi et al. [2010]. However, their methods incur still $\mathcal{O}(t)$ time and space complexities per round. Our goal is to design a method with low computation cost while maintaining strong regret guarantees.

**Remark 1** (*Unbounded GLMs*). Our GLM assumptions are aligned with the recent work [Lee et al., 2024], which are more general than the canonical GLM formulation introduced in Filippi et al. [2010] and later adopted in Sawarni et al. [2024]. Besides Assumptions 1 and 2, Filippi et al. [2010] further requires the rewards to be almost surely bounded, which automatically implies self-concordance and thus satisfies Assumption 3 as shown by Sawarni et al. [2024, Lemma 2.2]. Beyond bounded distributions, our GLM formulation accommodates unbounded ones, such as Gaussian or Poisson.◆

# 3 Proposed Method

This section presents our method for GLBs based on the principle of optimism in the face uncertainty (OFU) [Agrawal, 1995]. The core of our approach is a tight confidence set built on an online estimator. We begin with a review of the OFU principle and then introduce our method.

## 3.1 OFU Principle and Computational Challenge

**OFU Principle.** The OFU principle provides a principle way to balance exploration and exploitation in bandits. At each iteration $t$, this approach maintains a confidence set $\mathcal{C}_t(\delta) \subset \mathbb{R}^d$ to account for the uncertainty arising from the stochasticity of the historical data, ensuring it contains $\theta_*$ with high probability. Using the confidence set, one can construct a UCB for each action $\mathbf{x} \in \mathcal{X}_t$ as $\texttt{UCB}(\mathbf{x}) = \max_{\theta \in \mathcal{C}_t(\delta)} \mu(\mathbf{x}^\top \theta)$ and select the arm by $X_t = \arg\max_{\mathbf{x} \in \mathcal{X}_t} \texttt{UCB}(\mathbf{x})$. A key ingredient of OFU-based methods is the design of the confidence set, as the regret bound typically scales with the "radius" of the set. A tighter confidence set generally leads to a stronger regret guarantee.

**Computational Challenge.** To the best of our knowledge, most existing OFU-based methods for GLBs rely on maximum likelihood estimation (MLE) to estimate $\theta_*$ and construct the confidence set. Specifically, the estimator is computed as

$$\theta_t^{\text{MLE}} = \underset{\theta \in \Theta'}{\arg\min} \sum_{s=1}^{t-1} \ell_s(\theta) + \lambda \|\theta\|_2^2, \tag{2}$$

where $\ell_t(\theta) \triangleq (m(X_t^\top \theta) - r_t \cdot X_t^\top \theta)/g(\tau)$ is the loss function and $\lambda > 0$ is the regularizer parameter. The MLE was first used in the classical solution [Filippi et al., 2010], yet the regret bound exhibited linear dependence on $\kappa$ due to a loose analysis. Subsequent work [Lee et al., 2024, Sawarni et al., 2024, Liu et al., 2024] provided refined analyses showing that MLE is statistically efficient, with its estimation error relative to $\theta_*$ being independent of $\kappa$. This property enables the construction of a confidence set $\mathcal{C}_t(\delta)$ with a $\kappa$-free diameter, yielding the improved regret bound.

However, despite the statistical efficiency of the MLE, it has high computation cost. The existing methods mentioned above use different choices of the feasible domain $\Theta'$, but in all cases, solving the optimization problem requires access to the entire historical dataset, resulting in $\mathcal{O}(t)$ space complexity. Moreover, there is generally no closed-form solution for (2); the problem is typically solved using gradient-based methods such as projected gradient descent or Newton's method, where each gradient computation requires at least $\mathcal{O}(t)$ time per iteration [Filippi et al., 2010]. Consequently, both the time and space complexities of the MLE grow linearly with the number of rounds $t$, making it unfavorable for online settings. In addition, Sawarni et al. [2024] set $\Theta' = \mathbb{R}^d$ in (2) and required an additional projection step to ensure that $\theta$ lies within the desired domain. This projection involves solving a non-convex optimization problem, which is even more time-consuming.

## 3.2 Jointly Efficient Method

The main contribution of this paper is a statistically efficient confidence set $\mathcal{C}_t^{\text{OL}}(\delta)$ constructed based on an online estimator $\theta_t$, which has $\kappa$-free estimation error with respect to the true parameter $\theta_*$ and can be computed with $\mathcal{O}(1)$ time and space complexities per round.

**Online Estimator.** Drawing inspiration from the study for logistic bandits [Faury et al., 2022, Zhang and Sugiyama, 2023], we use the online mirror descent to learn the parameter $\theta_*$. For $t = 1$, we initialize $\theta_1 \in \Theta$ as any point in $\Theta$ and set $H_1 = \lambda I_d$. For time $t \geq 1$, we update the model by

$$\theta_{t+1} = \underset{\theta \in \Theta}{\arg\min} \, \widetilde{\ell}_t(\theta) + \frac{1}{2\eta} \|\theta - \theta_t\|_{H_t}^2, \tag{3}$$

where $\Theta = \{\theta \in \mathbb{R}^d \mid \|\theta\|_2 \leq S\}$ is a $d$-dimensional ball with radius $S$. In the above, we defined

$$\widetilde{\ell}_t(\theta) \triangleq \langle \nabla \ell_t(\theta_t), \theta - \theta_t \rangle + \frac{1}{2} \|\theta - \theta_t\|_{\nabla^2 \ell_t(\theta_t)}^2 \quad \text{and} \quad H_t \triangleq \lambda I_d + \sum_{s=1}^{t-1} \nabla^2 \ell_s(\theta_{s+1}). \tag{4}$$

The above two components play important roles in achieving low computation cost while maintaining statistical efficiency. The loss function $\widetilde{\ell}_t(\theta)$ serves as a second-order approximation of the original loss, which preserves the curvature information of the current loss function while ensures that the resulting optimization problem can be solved efficiently. The local matrix can also be expressed as $H_t = \lambda I_d + \sum_{s=1}^{t-1} \mu'(X_s^\top \theta_{s+1})/g(\tau) \cdot X_s X_s^\top$, where $X_s \in \mathbb{R}^d$ is the action selected by the learner. The matrix explicitly captures the non-linearity of the link function by $\mu'(X_s^\top \theta_{s+1})$ and retains the curvature information of historical loss functions until time $t - 1$. Since the optimization problem (3) is *quadratic optimization* over an Euclidean ball, it can be solved with a computation cost of $\mathcal{O}(d^3)$, independent of time $t$. Further details on solving (3) are provided in Appendix A.3.

**Confidence Set Construction.** Then, we can construct a tight confidence set based on the online estimator by carefully configuring the parameters as the following theorem.

**Theorem 1.** *Let $\delta \in (0, 1]$. Set the step size to $\eta = 1 + RS$ and the regularization parameter to $\lambda = \max\{14d\eta R^2, 6\eta RSL_\mu/g(\tau)\}$. For each $t \in [T]$, define the confidence set as*

$$\mathcal{C}_t^{\text{OL}}(\delta) \triangleq \{\theta \mid \|\theta - \theta_t\|_{H_t} \leq \beta_t(\delta)\}, \tag{5}$$

**Algorithm 1** GLB-OMD

---

**Input:** Self-concordant constant $R$, Lipchitz constant $L_\mu$, parameter radius $S$, confidence level $\delta$.

1: Initialize $\theta_1 \in \Theta := \{\theta \in \mathbb{R}^d \mid \|\theta\|_2 \leq S\}$ and $H_1 = \lambda I_d$.
2: **for** $t = 1$ to $T$ **do**
3:    Construct the confidence set $\mathcal{C}_t(\delta)$ according to (5).
4:    Select the arm $X_t$ according to rule (6) and receive the reward $r_t$.
5:    Update the online estimator $\theta_{t+1}$ by (3) and set $H_{t+1} = H_t + \nabla^2 \ell_t(\theta_{t+1})$.
6: **end for**

---

*where $\theta_t$ is the online estimator (3) and the radius $\beta_t(\delta)$ is given by*

$$\beta_t(\delta) = \sqrt{4\lambda S^2 + 2\eta \ln \frac{1}{\delta} + d(6\eta^2 + \eta) \ln\Big(1 + \frac{L_\mu}{\lambda g(\tau)}\Big)} = \mathcal{O}\left(SR\sqrt{d\Big(S^2 R + \ln \frac{t}{\delta}\Big)}\right).$$

*Then, under Assumptions 1, 2, and 3, we have $\Pr\Big[\forall t \geq 1, \; \theta_* \in \mathcal{C}_t(\delta)\Big] \geq 1 - \delta$. Besides, the time complexity for solving (3) is $\mathcal{O}(d^3)$, and the space complexity is $\mathcal{O}(d^2)$.*

Theorem 1 shows that an ellipsoidal confidence set $\mathcal{C}_t(\delta)$ can be constructed to quantify the uncertainty of the online estimator $\theta_t$ with both statistical and computational efficiency. From a computational perspective, constructing the confidence set relies only on the online estimator, which can be updated with $\mathcal{O}(1)$ time and space complexities. From a statistical view, the radius of the confidence set is independent of $\kappa$, which is crucial for achieving the improved regret bound, as detailed next.

**Remark 2** (Comparison with Logistic Bandits Literature). We note that OMD has been used in logistic bandits for constructing confidence sets [Faury et al., 2022, Zhang and Sugiyama, 2023, Lee and Oh, 2024]. However, existing methods are not fully jointly efficient compared with our result. Specifically, Faury et al. [2022] required optimization over the original loss at each round, incurring an additional $\mathcal{O}(\log t)$ computation cost and relying on a warm-up strategy to maintain statistical efficiency. In later work, Zhang and Sugiyama [2023] and Lee and Oh [2024] achieved a constant per-round cost but their regret bounds have an extra $\mathcal{O}(\sqrt{\log t})$ multiplicative factor. *More importantly*, as we will discuss in detail in Section 4, the analyses in these works depend on the specific structure of the logistic model and do not naturally extend to the GLB setting. Our key contribution is to introduce *mix-loss*-based technique into the confidence set analysis, which not only enables the application of OMD to the broader GLB framework but also improves both statistical and computational efficiency over previous methods for logistic bandits. ◆

**Arm Selection and Regret Bound.** Based on the ellipsoidal confidence set, one can employ a variety of exploration strategies, not limited to OFU-based methods but also including randomized approaches such as Thompson sampling [Abeille and Lazaric, 2017] for action selection. Here, we adopt the classical OFU-based strategy, where the action $X_t$ is selected by solving the bilevel optimization problem $X_t = \arg\max_{\mathbf{x} \in \mathcal{X}_t} \max_{\theta \in \mathcal{C}_t(\delta)} \mu(\mathbf{x}^\top \theta)$. Since $\mu$ is an increasing function and $\mathcal{C}_t(\delta)$ is an ellipsoid, the OFU-based action selection rule is equivalent to

$$X_t = \arg\max_{\mathbf{x} \in \mathcal{X}_t} \max_{\theta \in \mathcal{C}_t(\delta)} \mathbf{x}^\top \theta = \arg\max_{\mathbf{x} \in \mathcal{X}_t} \{\mathbf{x}^\top \theta_t + \beta_t(\delta) \|\mathbf{x}\|_{H_t^{-1}}\}, \tag{6}$$

which allows us to avoid solving the inner optimization problem explicitly. The overall implementation of the algorithm is summarized in Algorithm 1 and we have the following regret guarantee.

**Theorem 2.** *Let $\delta \in (0, 1]$. Under Assumptions 1, 2, and 3, with probability at least $1 - \delta$, Algorithm 1 with parameter $\eta = 1 + RS$ and $\lambda = \max\{14d\eta R^2, 6\eta RSL_\mu/g(\tau)\}$ ensures*

$$\text{REG}_T \lesssim dSR\sqrt{S^2 R + \log T}\sqrt{\frac{T \log T}{\kappa_*}} + \kappa d^2 S^2 R^3 \log T(S^2 R + \log T),$$

*where $\lesssim$ is used to hide constant independence of $d$, $\kappa$, $S$, $R$ and $T$.*

Theorem 2 shows that our method achieves an $\widetilde{\mathcal{O}}(d\sqrt{T/\kappa_*})$ regret, improving upon the $\widetilde{\mathcal{O}}(\kappa d\sqrt{T})$ bound of Filippi et al. [2010]. From a computational perspective, our OMD estimator can be updated in $\mathcal{O}(1)$ time and memory per round, in the same spirit as the least-squares estimator in linear bandits. Consequently, the computation cost of our algorithm matches that of LinUCB [Abbasi-Yadkori et al., 2011], indicating that the nonlinearity of the link function does not necessarily make GLBs more computationally demanding. In the finite-arm setting, our algorithm enjoys a constant per-round computational cost of $\mathcal{O}(d^3 + d^2|\mathcal{X}_t|)$, independent of $T$. In the infinite-arm case, the arm-selection step (6) could become the main computational bottleneck (once the computational issue of MLE is resolved). Our estimator remains broadly useful as a plug-in component that can be integrated into other exploration strategies, e.g., Thompson Sampling [Faury et al., 2022, Appendix D.2] where the arm-selection step reduces to a convex optimization problem for convex arm sets.

Since logistic bandits are a special case of GLBs, Theorem 2 also advances state-of-the-art results of logistic bandits by either reducing the $\mathcal{O}(\log t)$ time complexity of Faury et al. [2022] to $\mathcal{O}(1)$, or achieving an $\mathcal{O}(\sqrt{\log T})$ improvement in the regret bound over Zhang and Sugiyama [2023].

**Comparison with Sawarni et al. [2024].** We note that Sawarni et al. [2024] also pursued computational efficiency, but their approach is conceptually orthogonal to ours. They reduce the computation cost by employing a rare-update strategy that limits the frequency of parameter updates. However, their method remains MLE-based and requires storing all historical data, resulting in a memory cost of $\mathcal{O}(t)$. Moreover, although the rare-update strategy yields an amortized per-round time complexity of $\mathcal{O}((\log t)^2)$ over $T$ rounds, it still incurs a worst-case time complexity of $\mathcal{O}(t)$ in certain rounds. In contrast, our method performs a one-pass update with $\mathcal{O}(1)$ time and space complexities per round.

**Discussion with Lee and Oh [2025a].** The work [Lee and Oh, 2025a] (v3 version, the latest one available before the NeurIPS submitted date) builds on the framework of Zhang and Sugiyama [2023] and also reports an $\mathcal{O}(\sqrt{\log T})$ improvement in multinomial logit (MNL) bandits, a different problem that nonetheless shares certain technical connections with GLBs. Their technique could potentially be adapted to logistic bandits for an $\mathcal{O}(\log T \sqrt{T/\kappa_*})$ bound with $\mathcal{O}(1)$ cost. However, we identify potential technical issues in the analysis. The argument relies on a condition for the normalization factor of the truncated Gaussian distribution (see Eq.(C.15) of their paper, as also restated in (43)), an assumption that warrants further examination. We provide a detailed discussions in Appendix D.

### 3.3 More Discussions and Limitations

**Dependence on $T$, $\kappa$ and $d$.** For the dominant term, our regret bound matches the best-known results for GLBs using the MLE [Lee et al., 2024, Sawarni et al., 2024], with respect to its dependence on $T$, $\kappa$, and $d$. In terms of the non-leading term, Sawarni et al. [2024] achieved a slightly tighter bound, as it scales with $\kappa_\mathcal{X} = 1/\inf_{\mathbf{x}\in\cup_{t=1}^T \mathcal{X}_t} \mu'(\mathbf{x}^\top\theta_*)$, a quantity that can be smaller than $\kappa$. In the logistic bandit case, the non-leading term is further improved to be geometry-aware, adapting more precisely to the structure of the action set [Abeille et al., 2021]. We conjecture that similar improvements in the non-leading term, matching the $\kappa_\mathcal{X}$ dependence, might be achievable by incorporating a warm-up strategy, such as Procedure 1 in [Faury et al., 2022] or Algorithm 2 in Sawarni et al. [2024] to shift the curvature term from $\mu'(X_t^\top\theta_t)$ to $\mu'(X_t^\top\theta_*)$. However, it remains unclear whether geometry-aware bounds, akin to those in Abeille et al. [2021], can be obtained for GLBs without using the MLE.

**Dependence on $S$ and $R$.** The MLE-based method completely remove the dependence on $S$ and $R$ in the leading term [Lee et al., 2024], whereas our method still exhibits an $S^2$-dependence due to the requirement of one-pass updates. For MNL bandits, Lee and Oh [2025a] showed that one can achieved improved dependence on $S$ by incorporating an adaptive warm-up procedure. It may be possible to extend their warm-up technique to GLBs for a similar improvement on $S$.

## 4 Analysis

This section sketches the proof of Theorem 1 and highlights the key technical contributions.

### 4.1 A General Recipe for OMD

To prove Theorem 1, it suffices to show that the estimation error $\|\theta_{t+1} - \theta_*\|_{H_t} \leq \beta_t(\delta)$. The analysis begins with the following lemma, which is commonly used for the convergence or regret analysis for

the OMD-type update [Chen and Teboulle, 1993, Orabona, 2019, Zhao et al., 2024]. Here, we show it can serve as a general recipe for analyzing the estimation error of the OMD estimator.

**Lemma 1.** *Let $f : \Theta \to \mathbb{R}$ be a convex function on a convex set $\Theta$ and $A \in \mathbb{R}^{d \times d}$ be a symmetric positive definite matrix. Then, the update $\theta_{t+1} = \arg\min_{\theta \in \Theta} f(\theta) + \frac{1}{2\eta}\|\theta - \theta_t\|_A^2$ satisfies*

$$\|\theta_{t+1} - u\|_A^2 \leq 2\eta\langle\nabla f(\theta_{t+1}), u - \theta_{t+1}\rangle + \|\theta_t - u\|_A^2 - \|\theta_t - \theta_{t+1}\|_A^2, \;\; \textit{for all} \;\; u \in \Theta. \quad (7)$$

The above lemma provides a pathway to relate the estimation error to the so-called "inverse regret". In particular, under our configuration where $f(\theta) = \widetilde{\ell}_t(\theta)$, $A = H_t$, and $u = \theta_*$, and with a suitable choice of parameters, Lemma 4 in Appendix A shows that the estimator by (3) satisfies

$$\|\theta_t - \theta_*\|_{H_t}^2 \leq \underbrace{2\eta\left(\sum_{s=1}^{t-1}\ell_s(\theta_*) - \sum_{s=1}^{t-1}\ell_s(\theta_{s+1})\right)}_{\texttt{inverse regret}} - \frac{2}{3}\sum_{s=1}^{t-1}\|\theta_s - \theta_{s+1}\|_{H_s}^2 + \|\theta_1 - \theta_*\|_{H_1}^2, \quad (8)$$

We note that, although the above inequality has also been shown in the logistic bandits literature [Faury et al., 2022, Zhang and Sugiyama, 2023, Lee and Oh, 2024], the proof of (8) via Lemma 7 has not been explicitly discussed. We fill this gap by explicitly establishing the connection in our analysis.

## 4.2 Analysis for the Inverse Regret

**Main Challenge.** The main challenge and technical contribution of this paper lies in upper bounding the inverse regret term. Although previous works [Faury et al., 2022, Zhang and Sugiyama, 2023] have provided valuable insights, their techniques are challenging to extend to the GLB setting and remain suboptimal even for logistic bandits. The main technical difficulty in bounding the inverse regret is that $\theta_{s+1}$ is itself a function of the past losses $\ell_s$, which prevents the direct application of standard martingale concentration inequalities. A common strategy in prior work is to introduce an intermediate term by virtually running a full-information online learning algorithm, whose estimator $\widetilde{\theta}_s$ only depends on information up to time $s - 1$, allowing the inverse regret to be decomposed as

$$\texttt{inverse regret} = \underbrace{\sum_{s=1}^{t-1}\ell_s(\theta_*) - \sum_{s=1}^{t-1}\ell_s(\widetilde{\theta}_s)}_{\texttt{term (a)}} + \underbrace{\sum_{s=1}^{t-1}\ell_s(\widetilde{\theta}_s) - \sum_{s=1}^{t-1}\ell_s(\theta_{s+1})}_{\texttt{term (b)}}. \quad (9)$$

Here, the intermediate term $\sum_{s=1}^{t-1}\ell_s(\widetilde{\theta}_s)$ can be chosen as the cumulative loss of any online algorithm. In the case of logistic bandits, it is natural to leverage algorithms developed for online logistic regression, a well-studied problem with many established methods. For example, Faury et al. [2022] adopted the ALLIO algorithm [Jézéquel et al., 2020], while Zhang and Sugiyama [2023] built on the method proposed by Foster et al. [2018] and required the mixbaility property of the logistic loss. In contrast, for the GLB setting, the structure of the link function $\mu$ varies significantly across models, and it remains unclear how to design a unified intermediate algorithm. Moreover, even in the logistic bandit setting, existing analyses remain suboptimal. Specifically, Faury et al. [2022, Eq. (7)] required an online warm-up phase to shrink the feasible domain in order to bound term (b). Later, Zhang and Sugiyama [2023] avoided this warm-up step but instead relies on clipping the online estimator (see Eq. (35) in their paper), which incurs an additional $\mathcal{O}(\sqrt{\log T})$ term in the estimation error bound.

**Our Solution.** Instead of introducing an intermediate term via virtually running an online algorithm, we propose an alternative decomposition based on the *mix loss*, which is defined as

$$m_s(P_s) = -\ln\left(\mathbb{E}_{\theta \sim P_s}\left[e^{-\ell_s(\theta)}\right]\right), \quad (10)$$

where $P_s$ is a probability distribution over $\mathbb{R}^d$ whose specific form will be chosen later. In general, several choices of $P_s$ are possible, and we select the one that best fits our algorithmic design. The mix loss has played a central role in the analysis of exponentially weighted methods in full-information online learning [Vovk, 2001, van der Hoeven et al., 2018]. In our analysis, the mix loss is instrumental in analyzing the inverse regret defined in (9). In particular, the decomposition based on the mix loss $m_s(P_s)$ offers a more general and analytically versatile formulation than $\ell_s(\widetilde{\theta}_s)$ used in (9). The former reduces to the latter when $P_s$ is chosen as a Dirac distribution. We have the following lemma to upper bound the inverse regret under this mix-loss–based decomposition.

**Lemma 2** (informal). *Let $\{P_s\}_{s=1}^{\infty}$ be a stochastic process such that $P_s$ is a distribution over $\mathbb{R}^d$ and only relies on information collected until time $s-1$. Then, for any $\delta \in (0,1]$, we have*

$$\Pr\left[\forall t \geq 1, \sum_{s=1}^{t-1} \ell_s(\theta_*) - \sum_{s=1}^{t-1} m_s(P_s) \leq \log \frac{1}{\delta}\right] \geq 1 - \delta.$$

Lemma 2 follows from Ville's inequality [Ville, 1939], also known as "no-hypercompression" inequality [Grünwald, 2007, Chapter 3]. It implies that term (a) in (9) is upper-bounded by $\log(1/\delta)$ when the mix loss is used as an intermediate term, significantly smaller than the $\mathcal{O}((\log t)^3)$ bound in Zhang and Sugiyama [2023, Lemma 12] that employs the online logistic regression method [Foster et al., 2018]. Notably, Lemma 2 allows flexible choices of $P_s$, enabling us to tailor $P_s$ to closely track the OMD estimator's behavior. The formal statement is provided in Lemma 5 of Appendix A.

To bound term (b), we need to select $P_s = \mathcal{N}(\theta_s, 3\eta H_s^{-1}/2)$ as a Gaussian distribution centered at $\theta_s$ with covariance $\frac{3}{2}\eta H_s^{-1}$ to approximate the OMD estimator (3). We have the following lemma.

**Lemma 3** (informal). *Under Assumptions 1, 2, and 3, and with a suitable choice of $\lambda$, we have*

$$\sum_{s=1}^{t} m_s(P_s) - \sum_{s=1}^{t} \ell_s(\theta_{s+1}) \leq \frac{1}{3\eta} \sum_{s=1}^{t} \|\theta_{s+1} - \theta_s\|_{H_s}^2 + d(3\eta + \frac{1}{2})\ln\left(1 + \frac{L_\mu t}{\lambda g(\tau)}\right),$$

*where $P_s = \mathcal{N}(\theta_s, 3\eta H_s^{-1}/2)$ is a Gaussian distribution.*

Lemma 3 establishes a connection between the mix loss and the the cumulative loss of the "look-ahead" OMD estimator, where the loss of $\theta_{s+1}$ is measured over the $\ell_s$. A similar result for the logistic loss can be extracted from the proof of Lemma 14 in Zhang and Sugiyama [2023]. Here, we generalize this result to the GLB setting. Moreover, our proof simplifies the analysis in [Zhang and Sugiyama, 2023] by noticing that the mix loss can be interpreted as the convex conjugate of the KL divergence. The formal statement and complete proof are provided in Lemma 6 in Appendix A.

**More Technical Comparisons.** Our analysis of the inverse regret is closely connected to prior work that bounds the cumulative negative log likelihood $L_t(\theta_*) = \sum_{s=1}^{t} \ell_s(\theta_*)$ using Ville's inequality, a technique that traces back to Robbins [1970]. Most existing approaches aim to bound $L_t(\theta_*)$ with the loss of the MLE estimator, whereas our analysis focuses on the OMD estimator. This fundamental difference leads to a distinct construction of the intermediate term used in the decomposition.

In the context of GLB, our Lemma 2 resembles Lemma 3.3 of Lee et al. [2024], as both apply Ville's inequality. However, their intermediate term $\mathbb{E}_{\theta \sim P_t}[L_t(\theta)]$ is built upon a distribution $P_t$ that is fixed across all individual functions $\{\ell_s\}_{s=1}^{t-1}$, making it naturally aligned with the MLE estimator but does not readily adapt to changing estimator. In contrast, our OMD estimator evolves with each individual function. Defining the intermediate term as the mix loss with time-varying $P_s$ thus provides the flexibility needed to track this changing comparator. Very recently, two concurrent works [Kirschner et al., 2025, Clerico et al., 2025] have also employed the mix loss to derive confidence sets. Our Lemma 2 corresponds to Proposition 2.1 of Clerico et al. [2025], and the mix loss aligns with the sequential likelihood mixing technique of Kirschner et al. [2025]. While their analyses are already applicable to GLBs, a key difference is that we use an ellipsoidal confidence set centered at the OMD estimator to ensure computational efficiency, whereas both works [Kirschner et al., 2025, Clerico et al., 2025] primarily analyze the negative log-likelihood-based set, which are tighter but substantially more computationally demanding for GLBs. This difference leads to a different specification of the mix loss and requires a tailored analysis. Specifically, in their setting, $P_s$ is defined as the output of a continuous Hedge method to track the MLE estimator (or any fixed comparator). In contrast, our analysis must dynamically track the OMD estimator, which motivates our choice of $P_s$ as a Gaussian distribution with the evolving OMD estimators as its mean. Lemma 3 is then used to quantify the gap.

## 5 Experiment

This section evaluates the proposed method on two representative GLB problems: logistic bandits ($\mu(z) = 1/(1 + e^{-z})$) with bounded rewards, and Poisson bandits ($\mu(z) = e^z$), which pose a distinct challenge as an unbounded GLB setting. We also conduct experiments on real data from the Covertype dataset [Blackard, 1998], with more detailed results provided in Appendix E.

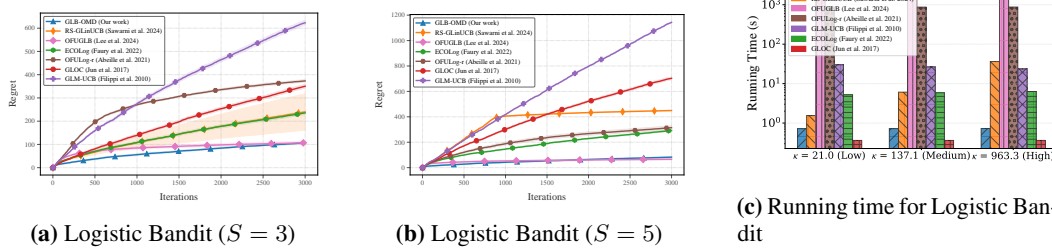

**(a)** Logistic Bandit ($S = 3$)   **(b)** Logistic Bandit ($S = 5$)   **(c)** Running time for Logistic Bandit

**Figure 1:** Regret and running time comparison of different algorithms on logistic bandits.

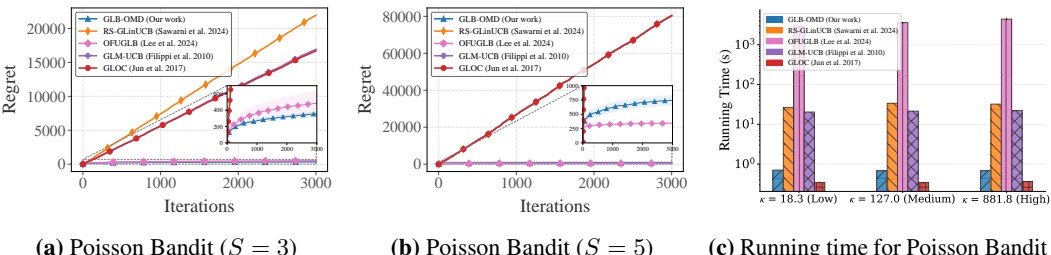

**(a)** Poisson Bandit ($S = 3$)   **(b)** Poisson Bandit ($S = 5$)   **(c)** Running time for Poisson Bandit

**Figure 2:** Regret and running time comparison of different algorithms on Poisson bandits.

**Compared Methods.** Four GLBs algorithms are compared, including `GLM-UCB` [Filippi et al., 2010], `GLOC` [Jun et al., 2017], `RS-GLinCB` [Sawarni et al., 2024], and `OFUGLB` [Lee et al., 2024]. For logistic bandits, we further include two specialized algorithms: an MLE-based method with nearly optimal regret `OFULog-r` [Abeille et al., 2021], and a jointly efficient method `ECOLog` [Faury et al., 2022]. We do not include `OFUL-MLogB` [Zhang and Sugiyama, 2023] since its confidence set is larger than that of ours, hence has a larger regret bound. More details of the baselines are provided in Appendix E. All experiments are conducted over 10 trials, and we report the average regret and running time.

**Results on Logistic Bandits.** We conduct experiments under different configurations of $S$. The underlying parameter $\theta_*$ is sampled from a $d$-dimensional sphere with radius $S = \{3, 5, 7\}$, corresponding to $\kappa = 21, 137,$ and $963$, respectively. Figure 1 reports the results. Among all methods, `GLOC` is the fastest but exhibits relatively large regret. `OFUGLB` attains the lowest regret due to its improved dependence on $\kappa$ and $S$, but as an MLE-based method, it incurs the highest computation cost. Our method strikes a favorable balance. Compared to `OFUGLB`, it achieves substantial cost savings with only a modest degradation in regret. Compared with `ECOLog` and `RS-GLinCB`, our method achieves comparable and even slightly better performance with improved computation cost. Moreover, it maintains a constant per-round cost across all regimes of $\kappa$, whereas the cost of `RS-GLinCB` increases with $\kappa$, as its rare-update strategy results in an update frequency that scales with $\kappa$.

**Results on Poisson Bandits.** We set the norm of the true parameter as $S \in \{3, 5, 7\}$, corresponding to $\kappa \approx 18, 127$ and $882$. Poisson bandits have unbounded rewards, whereas `GLM-UCB` and `RS-GLinCB` require a predefined upper bound on the maximum reward as a parameter. We set it as 100, assuming rewards are effectively bounded with high probability. As shown in Figure 2, our method reduces the computational cost of `OFUGLB` by roughly 1000 times, with only a modest increase in regret.

## 6 Conclusion

This paper proposed a new method for the GLB problem that achieves a nearly optimal regret bound of $\mathcal{O}(\log T \sqrt{T/\kappa_*})$ with $\mathcal{O}(1)$ time and space complexities per round. Our approach builds on a novel analysis of the OMD estimator using the mix loss, enabling a tight confidence set construction for arm selection. A natural extension is to incorporate a warm-up strategy, as in prior work, to improve the dependence on $S$ and obtain $\kappa_{\mathcal{X}}$-based bounds. It also remains open whether geometry-aware bounds for GLBs can be achieved similar to those in logistic bandits [Abeille et al., 2021]. Other directions include relaxing the self-concordance assumption toward weaker conditions [Liu et al., 2024], or improving $d$-dependence in the finite-arm setting [Jun et al., 2021, Mason et al., 2022].

## Acknowledgments and Disclosure of Funding

Peng Zhao was supported by NSFC (62206125) and the Xiaomi Foundation. MS was supported by the Institute for AI and Beyond, UTokyo. The authors thank the reviewers for their valuable feedback and for bringing to our attention the recent works [Kirschner et al., 2025, Clerico et al., 2025].

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

# A  Proof of Theorem 1

This section presents the proof of Theorem 1. We first provide the main proof, followed by the key lemmas used in the proof.

## A.1  Main Proof

*Proof of Theorem 1.* This part provides the proof of Theorem 1. By Lemma 4, when setting $\eta = 1 + RS$, we can upper bound the estimation error of the online estimator by the "inverse regret":

$$\|\theta_{t+1} - \theta_*\|_{H_t}^2 \leq 2\eta \underbrace{\left( \sum_{s=1}^{t} \ell_t(\theta_*) - \sum_{s=1}^{t} \ell_s(\theta_{s+1}) \right)}_{\texttt{inverse regret}} + 4\lambda S^2$$

$$+ \frac{2\eta RSL_\mu}{g(\tau)} \sum_{s=1}^{t} \|\theta_s - \theta_{s+1}\|_2^2 - \sum_{s=1}^{t} \|\theta_s - \theta_{s+1}\|_{H_s}^2. \tag{11}$$

Then, we can further decompose the "inverse regret term" into two parts:

$$\sum_{s=t}^{t} \ell_t(\theta_*) - \sum_{s=1}^{t} \ell_s(\theta_{s+1}) = \underbrace{\sum_{s=1}^{t} \ell_s(\theta_*) - \sum_{s=1}^{t} m_s(P_s)}_{\texttt{term (a)}} + \underbrace{\sum_{s=1}^{t} m_s(P_s) - \sum_{s=1}^{t} \ell_s(\theta_{s+1})}_{\texttt{term (b)}}. \tag{12}$$

In the above, we define $P_s = \mathcal{N}(\theta_s, \alpha H_s^{-1})$ as a $d$-dimensional multivariate Gaussian distribution with mean $\theta_s \in \mathbb{R}^d$ and covariance matrix $cH_s^{-1} \in \mathbb{R}^{d \times d}$, where $\alpha > 0$ is a constant to be specified latter. The function $m_s : P \mapsto \mathbb{R}$ that maps the distribution $P$ to a real number value is defined by

$$m_s(P_s) = -\ln \left( \mathbb{E}_{\theta \sim P_s} \left[ \exp \left( -\ell_s(\theta) \right) \right] \right).$$

We refer to the function $m_s$ as the "mix loss" because it mixes the loss with respect to the distribution $P_s$. This mixing has been found useful for achieving fast rates in prediction with expert advice and online optimization problems [Vovk, 2001]. Here, we show that the mix loss plays a crucial role in obtaining a jointly efficient online confidence set.

Given $P_s$ is a Gaussian distribution with mean $\theta_s$ and $\alpha H_s^{-1}$, it is $\mathcal{F}_s$-measurable. Then, Lemma 5 implies that for any $\delta \in (0, 1]$, we have

$$\texttt{term (a)} \leq \log \left( \frac{1}{\delta} \right), \tag{13}$$

with probability at least $1 - \delta$. Next, we proceed to analyze $\texttt{term (b)}$. Under the condition that $\lambda \geq 14 d \eta R^2$, Lemma 6 with $\alpha = \frac{3}{2}\eta$ shows that

$$\texttt{term (b)} \leq \frac{1}{3\eta} \sum_{s=1}^{t} \|\theta_{s+1} - \theta_s\|_{H_s}^2 + d(3\eta + \frac{1}{2}) \ln \left( 1 + \frac{L_\mu t}{\lambda g(\tau)} \right). \tag{14}$$

Plugging (12), (13), (14) into (11), we obtain

$$\|\theta_{t+1} - \theta_*\|_{H_t}^2 \leq 4\lambda S^2 + 2\eta \log \left( \frac{1}{\delta} \right) + d\eta(6\eta + 1) \ln \left( 1 + \frac{L_\mu t}{\lambda g(\tau)} \right)$$

$$+ \frac{2\eta RSL_\mu}{g(\tau)} \sum_{s=1}^{t} \|\theta_s - \theta_{s+1}\|_2^2 - \frac{1}{3} \sum_{s=1}^{t} \|\theta_s - \theta_{s+1}\|_{H_s}^2$$

$$\leq 4\lambda S^2 + 2\eta \log \left( \frac{1}{\delta} \right) + d\eta(6\eta + 1) \ln \left( 1 + \frac{L_\mu t}{\lambda g(\tau)} \right),$$

where the last inequality holds due to the condition $\lambda \geq 6\eta RSL_\mu/g(\tau)$. $\qquad\square$

## A.2 Useful Lemmas

This section presents several key lemmas used in the proof of Theorem 1.

**Lemma 4.** *Under Assumption 1 and 3 and setting $\eta = 1 + RS$, then for any $\lambda > 0$, the online estimator returned by (3) satisfies*

$$\|\theta_{t+1} - \theta_*\|_{H_{t+1}}^2 \leq (2 + 2RS)\left(\sum_{s=1}^{t} \ell_t(\theta_*) - \sum_{s=1}^{t} \ell_s(\theta_{s+1})\right) + 4\lambda S^2$$

$$+ \frac{L_\mu(2 + 2RS)}{g(\tau)}\sum_{s=1}^{t}\|\theta_s - \theta_{s+1}\|_2^2 - \sum_{s=1}^{t}\|\theta_s - \theta_{s+1}\|_{H_s}^2.$$

*Furthermore, if $\lambda \geq 6\eta L_\mu RS/g(\tau)$, we can further have*

$$\|\theta_{t+1} - \theta_*\|_{H_{t+1}}^2 \leq (2 + 2RS)\left(\sum_{s=1}^{t} \ell_t(\theta_*) - \sum_{s=1}^{t} \ell_s(\theta_{s+1})\right) - \frac{2}{3}\sum_{s=1}^{t}\|\theta_s - \theta_{s+1}\|_{H_s}^2 + 4\lambda S^2.$$

*Proof of Lemma 4.* We begin by using the integral formulation of Taylor's expansion. Since $\mu$ is twice differentiable, we have

$$\ell_s(\theta_{s+1}) - \ell_s(\theta_*) = \langle \nabla\ell_s(\theta_{s+1}), \theta_{s+1} - \theta_*\rangle - \|\theta_{s+1} - \theta_*\|_{\widetilde{h}_s}^2, \tag{15}$$

where $\widetilde{h}_s = \int_{v=0}^{1}(1-v)\nabla^2\ell_s(\theta_{s+1} + v(\theta_* - \theta_{s+1}))\,\mathrm{d}v$. By the definition of the loss function in (4), we can further express the Hessian as

$$\widetilde{h}_s = \frac{\widetilde{\alpha}(\theta_{s+1}, \theta_*, X_s)}{g(\tau)}X_s X_s^\top,$$

where $\widetilde{\alpha}(\theta_1, \theta_2, X_s) = \int_0^1(1-v)\,\mu'\left(X_s^\top\theta_1 + v\,X_s^\top(\theta_2 - \theta_1)\right)\mathrm{d}v$.

Next, under Assumption 3, we have $|\mu''(z)| \leq R\cdot\mu'(z)$ for all $z \in [-S, S]$. Consequently, Lemma 8 in Appendix C implies that for any $\theta_* \in \Theta \triangleq \{\theta \in \mathbb{R}^d \mid \|\theta\|_2 \leq S\}$,

$$\widetilde{\alpha}(\theta_{s+1}, \theta_*, X_s) \geq \frac{\mu'(X_s^\top\theta_{s+1})}{2 + 2RS}.$$

This inequality shows that

$$\widetilde{h}_s \succcurlyeq \frac{1}{2 + 2RS}\nabla^2\ell_s(\theta_{s+1}), \tag{16}$$

indicating that the Hessian $\widetilde{h}_s$ is bounded from below by $\frac{1}{2+2RS}\nabla^2\ell_s(\theta_{s+1})$ in the positive semidefinite order. Substituting (16) into (15) yields

$$\ell_s(\theta_{s+1}) - \ell_s(\theta_*) \leq \langle\nabla\ell_s(\theta_{s+1}), \theta_{s+1} - \theta_*\rangle - \frac{1}{2 + 2RS}\|\theta_{s+1} - \theta_*\|_{\nabla^2\ell_s(\theta_{s+1})}^2 \tag{17}$$

Since $\theta_{s+1}$ is the optimal solution of (3), Lemma 1 with $\mathbf{u} = \theta_*$ implies

$$\langle\nabla\ell_s(\theta_{s+1}), \theta_{s+1} - \theta_*\rangle$$
$$= \langle\nabla\ell_s(\theta_{s+1}) - \nabla\widetilde{\ell}_s(\theta_{s+1}), \theta_{s+1} - \theta_*\rangle + \langle\nabla\widetilde{\ell}_s(\theta_{s+1}), \theta_{s+1} - \theta_*\rangle$$
$$\leq \langle\nabla\ell_s(\theta_{s+1}) - \nabla\widetilde{\ell}_s(\theta_{s+1}), \theta_{s+1} - \theta_*\rangle$$
$$+ \frac{1}{2\eta}\left(\|\theta_s - \theta_*\|_{H_s}^2 - \|\theta_{s+1} - \theta_*\|_{H_s}^2 - \|\theta_s - \theta_{s+1}\|_{H_s}^2\right). \tag{18}$$

We can further express the first term on the right-hand side of (18) as

$$\langle \nabla \ell_s(\theta_{s+1}) - \nabla \widetilde{\ell}_s(\theta_{s+1}), \theta_{s+1} - \theta_* \rangle$$

$$= \langle \nabla \ell_s(\theta_{s+1}) - \nabla \ell_s(\theta_s) - \nabla^2 \ell_s(\theta_s)(\theta_{s+1} - \theta_s), \theta_{s+1} - \theta_* \rangle$$

$$= \frac{1}{g(\tau)} \langle \mu(X_s^\top \theta_{s+1}) \cdot X_s - \mu(X_s^\top \theta_s) \cdot X_s - \mu'(X_s^\top \theta_s) \cdot X_s X_s^\top (\theta_{s+1} - \theta_s), \theta_{s+1} - \theta_* \rangle$$

$$= \frac{\mu''(X_s^\top \boldsymbol{\xi}_s)}{2g(\tau)} \cdot \|\theta_s - \theta_{s+1}\|_{X_s X_s^\top}^2 \cdot X_s^\top (\theta_{s+1} - \theta_*)$$

$$\leq \frac{R}{2g(\tau)} \cdot \|\theta_s - \theta_{s+1}\|_{\mu'(X_s^\top \boldsymbol{\xi}_s) X_s X_s^\top}^2 \cdot X_s^\top (\theta_{s+1} - \theta_*)$$

$$\leq RS \|\theta_s - \theta_{s+1}\|_{\nabla^2 \ell_s(\boldsymbol{\xi}_s)}^2$$

$$\leq \frac{RSL_\mu}{g(\tau)} \|\theta_s - \theta_{s+1}\|_2^2, \tag{19}$$

where $\boldsymbol{\xi}_s \in \Theta$ lies on the line connecting $\theta_s$ and $\theta_{s+1}$. The first equality follows from the definition of $\nabla \widetilde{\ell}_s(\theta_{s+1})$ and the Taylor expansion with Lagrange's remainder. The first inequality uses the self-concordance property of the loss function, which ensures that $\mu''(z) \leq R\mu'(z)$ and $\mu'(z) \leq L_\mu$ for all $z \in [-S, S]$. The last inequality is due to $\|X_s\|_2 \leq 1$ and $\|\theta_s - \theta_{s+1}\|_2 \leq 2S$.

Combining (18), (19) with (17), setting $\eta = 1 + RS$ and taking the summation over $t \in [T]$ yield

$$\sum_{s=1}^{t} \ell_s(\theta_{s+1}) - \sum_{s=1}^{t} \ell_s(\theta_*)$$

$$\leq \frac{1}{2 + 2RS} \left( \|\theta_1 - \theta_*\|_{H_1}^2 - \|\theta_{t+1} - \theta_*\|_{H_{t+1}}^2 - \sum_{s=1}^{t} \|\theta_s - \theta_{s+1}\|_{H_s}^2 \right) + \frac{RSL_\mu}{g(\tau)} \|\theta_s - \theta_{s+1}\|_2^2.$$

We complete the proof by rearranging the terms and noticing $\|\theta_1 - \theta_*\|_{H_1}^2 \leq 4\lambda S^2$. $\qquad\square$

**Lemma 5.** *Let $\{\mathcal{F}_t\}_{t=1}^\infty$ be a filtration defined by $\mathcal{F}_t = \sigma\big(\{(X_s, r_s)\}_{s=1}^{t-1}\big)$. Let $\{P_t\}_{t=1}^\infty$ be a stochastic process such that the random variable $P_t$ is a distribution over $\mathbb{R}^d$ and is $\mathcal{F}_t$-measurable. Moreover, assume that the loss function $\ell_t$ defined by (4) is $\mathcal{F}_{t+1}$-measurable. For any $t \geq 1$, define*

$$L_t(\theta_*) = \sum_{s=1}^{t} \ell_s(\theta_*) \quad \text{and} \quad F_t = -\sum_{s=1}^{t} \ln \left( \mathbb{E}_{\theta \sim P_s} \left[ e^{-\ell_s(\theta)} \right] \right).$$

*Then, for any $\delta \in (0, 1]$, we have*

$$\Pr \left[ \forall t \geq 1, L_t(\theta_*) \leq F_t + \log\left(\frac{1}{\delta}\right) \right] \geq 1 - \delta.$$

*Proof of Lemma 5.* Let $M_0 = 1$ and for any $t \geq 1$, we define

$$M_t = \exp(L_t(\theta_*) - F_t).$$

To prove the lemma, it suffices to show that the sequence $\{M_t\}_{t=1}^\infty$ is a non-negative (super)-martingale; then, the maximum inequality Lattimore and Szepesvári [2020, Theorem 3.9] can be applied to obtain the desired result. To verify that $\{M_t\}_{t=1}^\infty$ is a (super)martingale, we begin by defining the density function of the natural exponential family distribution as follows:

$$p(r|z) = \exp \left( \frac{rz - m(z)}{g(\tau)} + h(r, \tau) \right),$$

where the function $m$, $g$ and $h$ share the same formulation as those in (1). Then, for each time $t \geq 1$, we can rewrite the expression of $M_t$ as

$$M_t = \frac{\prod_{s=1}^{t} E_{\theta \sim P_s} \left[ \exp(-\ell_s(\theta)) \right]}{\prod_{s=1}^{t} \exp\left(-\ell_s(\theta_*)\right)} = M_{t-1} \cdot \frac{\mathbb{E}_{\theta \sim P_t} \left[ \exp(-\ell_t(\theta)) \right]}{\exp\left(-\ell_t(\theta_*)\right)} = M_{t-1} \cdot \frac{\mathbb{E}_{\theta \sim P_t} [p(r_t|X_t^\top \theta)]}{p(r_t|X_t^\top \theta_*)}, \tag{20}$$

where the final equality holds because the loss function in (4) is the negative log-likelihood of an exponential family distribution; that is, for any $\theta \in \mathbb{R}^d$:

$$\exp\big(-\ell_t(\theta)\big) = \exp\left(\frac{r_t \cdot X_t^\top \theta - m(X_t^\top \theta)}{g(\tau)}\right) = p(r_t | X_t^\top \theta) \cdot \exp\big(-h(r_t, \tau)\big).$$

Then, by taking the conditional expectation with respect to the randomness in $r_t$ given $\mathcal{F}_t$ on both sides, we obtain:

$$
\begin{aligned}
\mathbb{E}[M_t | \mathcal{F}_t] &= M_{t-1} \cdot \mathbb{E}\left[\frac{\mathbb{E}_{\theta \sim P_t}[p(r_t | X_t^\top \theta)]}{p(r_t | X_t^\top \theta_*)} \,\Big|\, \mathcal{F}_t\right] \\
&= M_{t-1} \cdot \int \frac{\mathbb{E}_{\theta \sim P_t}[p(r | X_t^\top \theta)]}{p(r | X_t^\top \theta_*)} \cdot p(r | X_t^\top \theta_*) \mathrm{d}r \\
&= M_{t-1} \cdot \int \mathbb{E}_{\theta \sim P_t}[p(r | X_t^\top \theta)] \mathrm{d}r \\
&= M_{t-1} \cdot \mathbb{E}_{\theta \sim P_t}\left[\int p(r | X_t^\top \theta) \mathrm{d}r\right] \\
&= M_{t-1},
\end{aligned}
$$

where the first equality follows from the fact that $M_{t-1}$ is $\mathcal{F}_t$-measurable. The second inequality holds because the reward is sampled from the exponential family distribution (1). The final equality is a consequence of the tower property of conditional expectation. We have thus shown that $\{M_t\}_{t=1}^\infty$ is a martingale, and therefore a super-martingale. Then, by applying the maximum inequality Lattimore and Szepesvári [2020, Theorem 3.9], restating as Lemma 9 in Appendix C, we have

$$\Pr\left[\sup_{t \in \mathbb{N}} L_t(\theta_*) - F_t \geq \log \frac{1}{\delta}\right] \leq \delta M_0 = \delta,$$

which completes the proof. $\qquad\square$

**Lemma 6.** *Under Assumption 1, 2 and 3, let $P_s = \mathcal{N}(\theta_s, \alpha H_s^{-1})$ be a Gaussian distribution with mean $\theta_s$ and covariance matrix $\alpha H_s^{-1}$, where $H_s = \lambda I_d + \sum_{\tau=1}^{s-1} \nabla^2 \ell_\tau(\theta_{\tau+1})$ and $\alpha$ is any positive constant. We denote by $\theta_s$ the model returned by (3). Then, setting $\lambda \geq 64 d \alpha R^2 / 7$ we have*

$$\sum_{s=1}^t m_s(P_s) \leq \sum_{s=1}^t \ell_s(\theta_{s+1}) + \frac{1}{2\alpha} \sum_{s=1}^t \|\theta_{s+1} - \theta_s\|_{H_s}^2 + d\left(2\alpha + \frac{1}{2}\right) \ln\left(1 + \frac{2L_\mu t}{\lambda g(\tau)}\right),$$

*where the mix loss is defined as $m_s(P_s) = -\ln\big(\mathbb{E}_{\theta \sim P_s}\big[\exp\big(-\ell_s(\theta)\big)\big]\big)$.*

*Proof of Lemma 6.* Our analysis begin with the observation that the mix loss is a convex conjugate of the KL divergence function. Then Lemma 12 in Appendix C shows that

$$m_s(P_s) = -\log\left(\mathbb{E}_{\theta \sim P_s}[e^{-\ell_s(\theta)}]\right) = \underbrace{\mathbb{E}_{\theta \sim Q_s}[\ell_s(\theta)]}_{\texttt{term (a)}} + \underbrace{\mathrm{KL}(Q_s \| P_s)}_{\texttt{term (b)}} - \mathrm{KL}(Q_s \| P_s^*) \qquad (21)$$

for any distribution $Q_s$ defined over $\mathbb{R}^d$, where $P_s^*(\theta) \propto P_s(\theta) \cdot e^{-\ell_s(\theta)}$ for all $\theta \in \mathbb{R}^d$. Here, we choose $Q_s = \mathcal{N}(\theta_{s+1}, \alpha H_{s+1}^{-1})$ as a Gaussian distribution with mean $\theta_{s+1}$ and covariance $\alpha H_{s+1}^{-1}$.

Analysis of Term (a). Since $Q_s$ is symmetric around $\theta_{s+1}$, we can express term (a) as

$$
\begin{aligned}
\texttt{term (a)} &= \mathbb{E}_{\theta \sim Q_s}[\ell_s(\theta_{s+1}) + \langle \nabla \ell_s(\theta_{s+1}), \theta - \theta_{s+1}\rangle] + \mathbb{E}_{\theta \sim Q_s}[\mathcal{D}_{\ell_s}(\theta, \theta_{s+1})] \\
&= \ell_s(\theta_{s+1}) + \mathbb{E}_{\theta \sim Q_s}[\mathcal{D}_{\ell_s}(\theta, \theta_{s+1})] \\
&\leq \ell_s(\theta_{s+1}) + 2\alpha\Big(\log \det(H_{s+1}) - \log \det(H_s)\Big), \qquad (22)
\end{aligned}
$$

where $\mathcal{D}_{\ell_s}(\theta, \theta_{s+1}) = \ell_s(\theta) - \ell_s(\theta_{s+1}) - \langle \nabla \ell_s(\theta_{s+1}), \theta - \theta_{s+1}\rangle$ is the Bregman divergence of $\ell_s$ between $\theta$ and $\theta_{s+1}$. In the above, the second equality follows from the definition of $Q_s$. The last line follows from Lemma 7 in Appendix C, since $\ell_s$ is self-concordant and the condition $\lambda \geq 64 d \alpha R^2 / 7$ holds.

Analysis of Term (b). Given $Q_s$ and $P_s$ are both Gaussian distributions, Lemma 13 shows that

$$\texttt{term (b)} = \frac{1}{2}\left(\log\det(H_{s+1}) - \log\det(H_s) + \text{Tr}\left(H_s H_{s+1}^{-1}\right) + \frac{\|\theta_s - \theta_{s+1}\|_{H_s}^2}{\alpha} - d\right)$$

$$\leq \frac{1}{2}\left(\log\det(H_{s+1}) - \log\det(H_s)\right) + \frac{1}{2\alpha}\|\theta_s - \theta_{s+1}\|_{H_s}^2. \tag{23}$$

Put All Together. Plugging (22) and (23) into (21) and summing over $T$ rounds, we obtain

$$\sum_{s=1}^t m_s(P_s) \leq \sum_{s=1}^t \ell_s(\theta_{s+1}) + \frac{1}{2\alpha}\sum_{s=1}^t \|\theta_{s+1} - \theta_s\|_{H_s}^2 + (2\alpha + \frac{1}{2})\ln\left(\frac{\det(H_{t+1})}{\det(\lambda I_d)}\right).$$

The determinant of the matrix $H_{t+1}$ can be further bounded by

$$\det(H_{t+1}) = \det\left(\lambda I_d + \sum_{s=1}^t \nabla^2\ell_s(\theta_{s+1})\right) \leq \det\left(\left(\lambda + L_\mu t/g(\tau)\right)I_d\right) \leq \left(\lambda + L_\mu t/g(\tau)\right)^d.$$

Then, we obtain

$$\sum_{s=1}^t m_s(P_s) \leq \sum_{s=1}^t \ell_s(\theta_{s+1}) + \frac{1}{2\alpha}\sum_{s=1}^t \|\theta_{s+1} - \theta_s\|_{H_s}^2 + d(2\alpha + \frac{1}{2})\ln\left(1 + \frac{L_\mu t}{\lambda g(\tau)}\right),$$

which completes the proof. $\qquad\square$

**Lemma 7.** *Let $\ell_s(\theta)$ be the loss function of the maximum likelihood estimator and $Q_s = \mathcal{N}(\theta_{s+1}, \alpha H_{s+1}^{-1})$ be a Gaussian distribution with mean $\theta_{s+1}$ and covariance matrix $\alpha H_{s+1}^{-1}$ with $H_s = \lambda I_d + \sum_{\tau=1}^{s-1}\nabla^2\ell_\tau(\theta_{\tau+1})$. Under Assumption 1, 2 and 3 and setting $\lambda \geq 64 d\alpha R^2/7$, for any constant $\alpha > 0$, we have*

$$\mathbb{E}_{\theta\sim Q_s}[\mathcal{D}_{\ell_s}(\theta, \theta_{s+1})] \leq 2\alpha\Big(\log det(H_{s+1}) - \log det(H_s)\Big),$$

*Proof of Lemma 7.* By the the integral formulation of Taylor's expansion and the definition of $\ell_s$ such that $\nabla^2\ell_s(\theta) = \mu'(X_s^\top\theta)/g(\tau)\cdot X_s X_s^\top$ for any $\theta\in\mathbb{R}^d$, we have

$$\mathbb{E}_{\theta\sim Q_s}[\mathcal{D}_{\ell_s}(\theta, \theta_{s+1})] = \mathbb{E}_{\theta\sim Q_s}\left[\|\theta - \theta_{s+1}\|_{\widetilde{h}_s(\theta)}^2\right],$$

where $\widetilde{h}_s(\theta) = \int_{v=0}^1 (1-v)\mu'\left(X_s^\top\theta_{s+1} + vX_s^\top(\theta - \theta_{s+1})\right)\mathrm{d}v\cdot X_s X_s^\top/g(\tau)$. According to Lemma 8 in Appendix C and the condition $\|X_t\|_2 \leq 1$ by Assumption 1, we can bound the Hessian $\widetilde{h}_s(\theta)$ by

$$\widetilde{h}_s(\theta) \leq \exp(R^2\|\theta - \theta_{s+1}\|_2^2)\cdot\nabla^2\ell_s(\theta_{s+1}). \tag{24}$$

Then, the approximation error between the linearized loss $g_s(\theta)$ and $\ell_s(\theta)$ is bounded by

$$\mathbb{E}_{\theta\sim Q_s}[\mathcal{D}_{\ell_s}(\theta, \theta_{s+1})] \leq \mathbb{E}_{\theta\sim Q_s}\left[e^{R^2\|\theta-\theta_{s+1}\|_2^2}\cdot\|\theta - \theta_{s+1}\|_{\nabla^2\ell_s(\theta_{s+1})}^2\right]$$

$$\leq \sqrt{\mathbb{E}_{\theta\sim Q_s}\left[e^{2R^2\|\theta-\theta_{s+1}\|_2^2}\right]\cdot\mathbb{E}_{\theta\sim Q_s}\left[\|\theta - \theta_{s+1}\|_{\nabla^2\ell_s(\theta_{s+1})}^4\right]}$$

$$\leq \sqrt{\frac{4}{3}\mathbb{E}_{\theta\sim Q_s}\left[\left\|\left(\nabla^2\ell_s(\theta_{s+1})\right)^{\frac{1}{2}}(\theta - \theta_{s+1})\right\|_2^4\right]}, \tag{25}$$

The second inequality is due to the Cauchy-Schwarz inequality. The last inequality is due to Lemma 10 under the condition that $H_s \succcurlyeq \lambda I_d$ and the setting $\lambda \geq 64 d\alpha R^2/7$. For the last term on the right hand side of (25), the random variable $\left(\nabla^2\ell_s(\theta_{s+1})\right)^{\frac{1}{2}}(\theta - \theta_{s+1})$ follows the same distribution as

$$\sum_{i=1}^d \sqrt{\alpha\lambda_i}X_i\mathbf{e}_i \quad \text{and} \quad X_i \overset{iid}{\sim} N(0, 1), \forall i\in[d],$$

where $\lambda_i$ is the $i$-th largest eigenvalue of the matrix $\left(\nabla^2\ell_s(\theta_{s+1})\right)^{\frac{1}{2}} H_{s+1}^{-1} \left(\nabla^2\ell_s(\theta_{s+1})\right)^{\frac{1}{2}}$ and $\mathbf{e}_i$ is a set of orthogonal basis. Then, we have

$$\sqrt{\mathbb{E}_{\theta\sim Q_s}\left[\left\|\left(\nabla^2\ell_s(\theta_{s+1})\right)^{\frac{1}{2}}(\theta-\theta_{s+1})\right\|_2^4\right]} = \sqrt{\mathbb{E}_{X_i\sim\mathcal{N}(0,1)}\left[\left(\sum_{i=1}^d \alpha\lambda_i X_i^2\right)^2\right]}$$

$$= \sqrt{\sum_{i=1}^d\sum_{j=1}^d \alpha^2\lambda_i\lambda_j \mathbb{E}_{X_i,X_j\sim\mathcal{N}(0,1)}[X_i^2 X_j^2]}$$

$$= \sqrt{3}\alpha\mathrm{Tr}\left(H_{s+1}^{-1}\left(\nabla^2\ell_s(\theta_{s+1})\right)\right). \tag{26}$$

where $\mathrm{Tr}(A)$ denotes the trace of matrix $A$. In the above, the last inequality is due to the fact that $\mathbb{E}_{X_i,X_j\sim\mathcal{N}(0,1)}[X_i X_j] = 3$. The last inequality is due to $\mathrm{trace}(AB) = \mathrm{trace}(BA)$ for matrix $A, B \in \mathbb{R}^{d\times d}$. Recall that $H_{s+1} = \lambda I_d + \sum_{\tau=1}^s \nabla^2\ell_\tau(\theta_{\tau+1})$.

$$\mathrm{Tr}\left(H_{s+1}^{-1}\left(\nabla^2\ell_s(\theta_{s+1})\right)\right) \leq \mathrm{Tr}\left(H_{s+1}^{-1}(H_{s+1} - H_s)\right) = \mathrm{Tr}\left(I - H_s H_{s+1}^{-1}\right)$$

$$\leq \log\det(H_{s+1}) - \log\det(H_s). \tag{27}$$

Combining (26) and (27) with (25) yields the desired result. $\qquad\square$

## A.3 Computational Cost Discussion

This part discusses the time and space complexity of solving the optimization problem (3).

**Proposition 1.** *The time complexity for solving (3) is $\mathcal{O}(d^3)$, and the space complexity is $\mathcal{O}(d^2)$.*

*Proof of Proposition 1.* We begin with the analysis on the time complexity, followed by the discussion on the space complexity.

Time Complexity. According to Theorem 6.23 of Orabona [2019], the update rule of online mirror descent (3) can be equivalently expressed as

$$\zeta_{t+1} = \theta_t - \eta\widetilde{H}_t^{-1}\nabla\ell_t(\theta_t), \tag{28a}$$

$$\theta_{t+1} = \arg\min_{\theta\in\Theta}\|\theta - \zeta_{t+1}\|_{\widetilde{H}_t}^2, \tag{28b}$$

where $\widetilde{H}_t = H_t + \eta\nabla^2\ell_t(\theta_t)$. In this formulation, the first step (28a) is a gradient update, whose main computational cost lies in computing the inverse of the Hessian matrix. Since $\nabla^2\ell_t(\theta_t) = \mu'(X_t^\top\theta_t)/g(\tau) \cdot X_t X_t^\top$ is a rank-1 matrix, the Sherman-Morrison formula can be applied to efficiently compute the inverse of $\widetilde{H}_t$ as

$$(\widetilde{H}_t + \eta\nabla^2\ell_t(\theta_t))^{-1} = H_t^{-1} - \frac{H_t^{-1} X_t X_t^\top H_t^{-1}}{\frac{g(\tau)}{\eta\mu'(X_t^\top\theta_t)} + X_t^\top H_t^{-1} X_t},$$

which reduces the computational complexity to $\mathcal{O}(d^2)$ per iteration, assuming $H_t^{-1}$ is available. Since $H_t = H_{t-1} + \nabla^2\ell_{t-1}(\theta_t)$, $H_t^{-1}$ can also be updated by the Sherman-Morrison formula in $\mathcal{O}(d^2)$ time per round based on $H_{t-1}^{-1}$. Therefore, the total computational cost of (28a) is $\mathcal{O}(d^2)$. In the second step (28b), as $\widetilde{H}_t$ is positive semi-definite, the optimization problem can be solved in $\mathcal{O}(d^3)$ time (see Section 4.1 of [Mhammedi et al., 2019] for details). Overall, the total time complexity for solving (3) is $\mathcal{O}(d^3)$.

Space Complexity. Regarding space complexity, it suffices to store the current model $\theta_t$, the gradient $\nabla\ell_t(\theta_t)$, the inverse Hessian matrix $H_t^{-1}$, and $\widetilde{H}_t^{-1}$ throughout the optimization process, resulting in a total space complexity of $\mathcal{O}(d^2)$. $\qquad\square$

# B  Proof of Theorem 2

*Proof of Theorem 2.* Let $(X_t, \widetilde{\theta}_t) = \arg\max_{\mathbf{x}\in\mathcal{X}_t, \theta\in\mathcal{C}_t(\delta)} \mu(\mathbf{x}^\top\theta)$. We can bound the regret by

$$
\begin{aligned}
\mathrm{REG}_T &= \sum_{t=1}^{T} \mu(\mathbf{x}_{t,*}^\top\theta_*) - \sum_{t=1}^{T}\mu(X_t^\top\theta_*) \\
&\leq \sum_{t=1}^{T}\mu(X_t^\top\widetilde{\theta}_t) - \sum_{t=1}^{T}\mu(X_t^\top\theta_*) \\
&= \underbrace{\sum_{t=1}^{T}\mu'(X_t^\top\theta_*)X_t^\top(\widetilde{\theta}_t - \theta_*)}_{\texttt{term (a)}} + \underbrace{\frac{1}{2}\sum_{t=1}^{T}\widetilde{\alpha}(\theta_*, \widetilde{\theta}_t, X_t)\big(X_t^\top(\widetilde{\theta}_t - \theta_*)\big)^2}_{\texttt{term (b)}},
\end{aligned} \qquad (29)
$$

where $\widetilde{\alpha}(\theta_1, \theta_2, X_s) = \int_0^1 (1-v)\mu''\big(X_s^\top\theta_1 + v\,X_s^\top(\theta_2 - \theta_1)\big)\,\mathrm{d}v$. In the above, the first inequality is due to the arm selection rule (6) and the second equality is by the integral formulation of the Taylor's expansion. Then, we upper bound the terms respectively.

**Analysis for term (a).**  For the first term, we have

$$
\begin{aligned}
\texttt{term (a)} &= \sum_{t=1}^{T}\mu'(X_t^\top\theta_*)\cdot X_t^\top(\widetilde{\theta}_t - \theta_*) \\
&\leq \sum_{t=1}^{T}\mu'(X_t^\top\theta_*)\cdot \|X_t\|_{H_t^{-1}}\|\widetilde{\theta}_t - \theta_*\|_{H_t} \\
&\leq 2\sum_{t=1}^{T}\mu'(X_t^\top\theta_*)\cdot \beta_t(\delta)\|X_t\|_{H_t^{-1}} \\
&\leq 2\beta_T(\delta)\sum_{t=1}^{T}\mu'(X_t^\top\theta_*)\cdot \|X_t\|_{H_t^{-1}} \\
&\leq \underbrace{2\beta_T(\delta)\sum_{t\in\mathcal{T}_1}\mu'(X_t^\top\theta_*)\cdot \|X_t\|_{H_t^{-1}}}_{\texttt{term (a1)}} + \underbrace{2\beta_T(\delta)\sum_{t\in\mathcal{T}_2}\mu'(X_t^\top\theta_*)\cdot \|X_t\|_{H_t^{-1}}}_{\texttt{term (a2)}},
\end{aligned}
$$

where the first inequality is due to the Hölder's inequality and the second inequality is by the fact $\|\widetilde{\theta}_t - \theta_*\|_{H_t} \leq \|\widetilde{\theta}_t - \theta_t\|_{H_t} + \|\theta_t - \theta_*\|_{H_t} \leq 2\beta_t(\delta)$. In the last inequality, we decompose the time horizon into $\mathcal{T}_1 = \{t\in[T]\mid \mu'(X_t^\top\theta_*) \geq \mu'(X_t^\top\theta_{t+1})\}$ and $\mathcal{T}_2 = [T]/\mathcal{T}_1$.

*Analysis for Term (a1)*: For the time steps in $t\in\mathcal{T}_1$, the term $\mu'(X_t^\top\theta_*)$ can be further bounded by

$$
\begin{aligned}
\mu'(X_t^\top\theta_*) &= \mu'(X_t^\top\theta_{t+1}) + \int_{v=0}^{1}\mu''\big(X_t^\top\theta_{t+1} + vX_t^\top(\theta_* - \theta_{t+1})\big)\mathrm{d}v \cdot X_t^\top(\theta_* - \theta_{t+1}) \\
&\leq \mu'(X_t^\top\theta_{t+1}) + R\int_{v=0}^{1}\mu'\big(X_t^\top\theta_{t+1} + vX_t^\top(\theta_* - \theta_{t+1})\big)\mathrm{d}v \cdot |X_t^\top(\theta_* - \theta_{t+1})| \\
&\leq \mu'(X_t^\top\theta_{t+1}) + RL_\mu\cdot\|X_t\|_{H_{t+1}^{-1}}\cdot\|\theta_* - \theta_{t+1}\|_{H_{t+1}} \\
&\leq \mu'(X_t^\top\theta_{t+1}) + RL_\mu\beta_{t+1}(\delta)\cdot\|X_t\|_{H_t^{-1}}
\end{aligned} \qquad (30)
$$

where the first inequality is due the self-concordant property of $\mu$. The second inequality is by Assumption 2 such that $\mu'(z) \leq L_\mu$ for $z\in[-S, S]$ and the Hölder's inequality. The last inequality is due to Theorem 1 and the fact $H_{t+1} \succcurlyeq H_t$.

Then, let $\widetilde{H}_t := g(\tau)H_t = \lambda g(\tau)I_d + \sum_{s=1}^{t-1} \mu'(X_s^\top \theta_{s+1})X_s X_s^\top$ and $V_t := \lambda g(\tau)I_d + \frac{1}{\kappa}\sum_{s=1}^{t-1} X_s X_s^\top$. We can upper term (a1) by

$$
\begin{aligned}
\texttt{term (a1)} &\leq \frac{2\beta_T(\delta)}{\sqrt{g(\tau)}} \sum_{t\in\mathcal{T}_1} \mu'(X_t^\top\theta_{t+1})\|X_t\|_{\widetilde{H}_t^{-1}} + \frac{2RL_\mu \beta_{T+1}^2(\delta)}{g(\tau)} \sum_{t=1}^{T}\|X_t\|_{\widetilde{H}_t^{-1}}^2 \\
&\leq \frac{2\beta_T(\delta)}{\sqrt{g(\tau)}} \sum_{t\in\mathcal{T}_1} \sqrt{\mu'(X_t^\top\theta_*)} \cdot \left\|\sqrt{\mu'(X_t^\top\theta_{t+1})}X_t\right\|_{\widetilde{H}_t^{-1}} + \frac{2RL_\mu \beta_{T+1}^2(\delta)}{g(\tau)} \sum_{t\in\mathcal{T}_1}\|X_t\|_{\widetilde{H}_t^{-1}}^2 \\
&\leq \frac{2\beta_T(\delta)}{\sqrt{g(\tau)}} \sqrt{\sum_{t\in\mathcal{T}_1} \mu'(X_t^\top\theta_*)} \cdot \sqrt{\sum_{t\in\mathcal{T}_1}\left\|\sqrt{\mu'(X_t^\top\theta_{t+1})}X_t\right\|_{\widetilde{H}_t^{-1}}^2} + \frac{2RL_\mu \beta_{T+1}^2(\delta)}{g(\tau)} \sum_{t\in\mathcal{T}_1}\|X_t\|_{V_t^{-1}}^2,
\end{aligned}
$$
(31)

where the first inequality is by the condition $\mu'(X_t^\top\theta_{t+1}) \leq \mu'(X_t^\top\theta_*)$ for $t \in \mathcal{T}_1$. The second inequality is due to the Cauchy-Schwarz inequality and the fact that $\widetilde{H}_t \succeq V_t$. Then, we can further bound (31) by elliptical potential lemma (Lemma 11 in Appendix C) as:

$$
\sum_{t\in\mathcal{T}_1}\left\|\sqrt{\mu'(X_t^\top\theta_{t+1})}X_t\right\|_{\widetilde{H}_t^{-1}}^2 \leq \sum_{t\in[T]}\left\|\sqrt{\mu'(X_t^\top\theta_{t+1})}X_t\right\|_{\widetilde{H}_t^{-1}}^2 \leq 2d(1+L_\mu)\log\left(1 + \frac{TL_\mu}{g(\tau)d\lambda}\right)
$$
(32)

by taking $\mathbf{z}_t = \sqrt{\mu'(X_t^\top\theta_{t+1})}X_t$ in Lemma 11. The last term in (31) can also be bounded by

$$
\sum_{t\in\mathcal{T}_1}\|X_t\|_{V_t^{-1}}^2 \leq \sum_{t\in[T]}\|X_t\|_{V_t^{-1}}^2 \leq 4d\log\left(1 + \frac{T}{g(\tau)\kappa d\lambda}\right).
$$
(33)

For notational simplicity, we denote by

$$
\begin{cases}
\gamma_T^{(1)}(\delta) = \frac{2\beta_T(\delta)\sqrt{2d(1+L_\mu)}}{\sqrt{g(\tau)}}\sqrt{\log\left(1 + \frac{TL_\mu}{g(\tau)d\lambda}\right)} = \mathcal{O}(\beta_T(\delta)\sqrt{d\log T}), \\
\gamma_T^{(2)}(\delta) = \frac{8\kappa dRL_\mu \beta_{T+1}^2(\delta)}{g(\tau)}\log\left(1 + \frac{T}{g(\tau)\kappa d\lambda}\right) = \mathcal{O}\left(\beta_{T+1}(\delta)^2\kappa dR\log T\right).
\end{cases}
$$
(34)

Then, plugging (32) and (33) into (31) yields

$$
\texttt{term (a1)} \leq \gamma_T^{(1)}(\delta)\sqrt{\sum_{t\in\mathcal{T}_1}\mu'(X_t^\top\theta_*)} + \gamma_T^{(2)}(\delta) \leq \gamma_T^{(1)}(\delta)\sqrt{T/\kappa_* + R\cdot\text{REG}_T} + \gamma_T^{(2)}(\delta), \quad (35)
$$

where the last inequality can be obtained following the same arguments in the proof of [Abeille et al., 2021, Theorem 1].

*Analysis for Term (a2)*: As for the term (a2), we have

$$
\texttt{term (a2)} \leq \beta_T(\delta)\sum_{t\in\mathcal{T}_2}\mu'(X_t^\top\theta_*)\cdot\|X_t\|_{H_t^{-1}} \leq \beta_T(\delta)\sum_{t\in\mathcal{T}_2}\sqrt{\mu'(X_t^\top\theta_*)}\cdot\left\|\sqrt{\mu'(X_t^\top\theta_{t+1})}X_t\right\|_{H_t^{-1}},
$$

where the last inequality holds due to the condition $\mu'(X_t^\top\theta_*) \leq \mu'(X_t^\top\theta_{t+1})$. Following the same arguments in bounding (31), we can obtain

$$
\texttt{term (a2)} \leq \gamma_T^{(1)}(\delta)\sqrt{T/\kappa_* + R\cdot\text{REG}_T}.
$$

Combining the upper bound for term (a1) and term (a2), we have

$$
\texttt{term (a)} \leq 2\gamma_T^{(1)}(\delta)\sqrt{T/\kappa_* + R\cdot\text{REG}_T} + \gamma_T^{(2)}(\delta).
$$

**Analysis for term (b).** As for the term by, we have

$$
\begin{aligned}
\texttt{term (b)} &= \frac{1}{2} \sum_{t=1}^{T} \widetilde{\alpha}(\theta_*, \widetilde{\theta}_t, X_t) \big(X_t^\top (\widetilde{\theta}_t - \theta_*)\big)^2 \\
&\leq \frac{R}{2} \sum_{t=1}^{T} \int_0^1 \mu'\big(X_s^\top \theta_* + v\, X_s^\top (\widetilde{\theta}_t - \theta_*)\big)\, \mathrm{d}v \big(X_t^\top (\widetilde{\theta}_t - \theta_*)\big)^2 \\
&\leq \frac{RL_\mu}{2} \sum_{t=1}^{T} \|\widetilde{\theta}_t - \theta_*\|_{H_t}^2 \cdot \|X_t\|_{H_t^{-1}}^2 \\
&\leq \frac{2RL_\mu \beta_T^2(\delta)}{g(\tau)} \sum_{t=1}^{T} \|X_t\|_{V_t^{-1}}^2 \leq \gamma_T^{(2)}(\delta),
\end{aligned}
\tag{36}
$$

where the first inequality is due to the self-concordant property of $\mu$. The second inequality is by Assumption 2 and Cauchy-Schwarz inequality. The third inequality is due to the fact $\|\widetilde{\theta}_t - \theta_*\|_{H_t}^2 \leq 4\beta_T^2(\delta)$ and $H_t \succcurlyeq g(\tau)V_t$. The last line can be obtained following the same arguments in bounding (33).

**Overall Regret Bound.** Plugging (35) and (36) into (29) yields

$$
\mathrm{REG}_T \leq 2\gamma_T^{(1)}(\delta)\sqrt{T/\kappa_* + R \cdot \mathrm{REG}_T} + 2\gamma_T^{(2)}(\delta).
$$

Removing the above inequality and rearranging the terms yields

$$
\begin{aligned}
\mathrm{REG}_T &\leq 2\gamma_2 + 2\gamma_1^2 R + 2\gamma_1 \sqrt{\gamma_1^2 R^2 + 2\gamma_2 R + T/\kappa_*} \\
&\leq 2\gamma_2 + 2\gamma_1^2 R + 2\gamma_1 \left(\gamma_1 R + \sqrt{2\gamma_2 R} + \sqrt{T/\kappa_*}\right) \\
&\leq 2\gamma_2 + 4\gamma_1^2 R + 2\gamma_1 \sqrt{2\gamma_2 R} + 2\gamma_1 \sqrt{T/\kappa_*} \\
&\leq \mathcal{O}\left(\beta_T(\delta)\sqrt{dT\log T/\kappa_*} + \kappa dR \log T \beta_T^2(\delta)\right) \\
&\leq \mathcal{O}\left(dSR\sqrt{S^2 R + \log T}\sqrt{\frac{T\log T}{\kappa_*}} + \kappa d^2 S^2 R^3 \log T(S^2 R + \log T)\right),
\end{aligned}
$$

where $\gamma_1 = \gamma_T^{(1)}(\delta)$ and $\gamma_2 = \gamma_T^{(2)}(\delta)$ is defined as (34). We have completed the proof of the regret.

**Computational Complexity.** As shown in Proposition 1 in Appendix A.3, the time complexity for updating the online estimator $\theta_t$ is $\mathcal{O}(d^3)$ (line 5 in Algorithm 1). Additionally, the inverse Hessian matrix $H_t^{-1}$ can be updated in $\mathcal{O}(d^2)$ time per round as shown in Appendix A.3. The remaining computational cost arises from the arm selection (6), which solves the optimization problem

$$
X_t = \underset{\mathbf{x} \in \mathcal{X}_t}{\arg\max}\{\mathbf{x}^\top \theta_t + \beta_t(\delta)\|\mathbf{x}\|_{H_t^{-1}}\}.
$$

Given $\theta_t$ and $H_t^{-1}$, this optimization can be performed in $\mathcal{O}(d^2|\mathcal{X}_t|)$ time at round $t$. Therefore, the total per-round computational complexity is $\mathcal{O}(d^3 + d^2|\mathcal{X}_t|)$.

$\square$

## C  Technical Lemmas

**Lemma 8** (Lemma 9 of Faury et al. [2020]). *Let $\mu : \mathbb{R} \to \mathbb{R}$ be a strictly increasing function satisfying $|\mu''(z)| \leq R\,\mu'(z)$ for all $z \in \mathcal{Z}$, where $R > 0$ is a fixed positive constant and $\mathcal{Z} \subset \mathbb{R}$ is a bounded interval. Then, for any $z_1, z_2 \in \mathcal{Z}$ and $z \in \{z_1, z_2\}$, we have*

$$
\int_{\nu=0}^1 \mu'(z_1 + \nu(z_2 - z_1))\mathrm{d}\nu \geq \frac{\mu'(z)}{1 + R \cdot |z_1 - z_2|}.
\tag{37}
$$

*and the weighted integral*

$$\frac{\mu'(z)}{2 + R \cdot |z_1 - z_2|} \leq \int_{\nu=0}^{1} (1 - \nu)\mu'(z_1 + \nu(z_2 - z_1))d\nu \leq \exp\left(R^2|z_1 - z_2|^2\right) \cdot \mu'(z). \quad (38)$$

We include the proof here for self-containedness.

*Proof of Lemmas 8.* Without loss of generality, assume $z = z_1$. Let $\phi(\nu) := \mu'(z_1 + \nu(z_2 - z_1))$ and $\Delta := |z_2 - z_1|$. From $|\mu''(z)| \leq R\mu'(z)$, we obtain the key differential inequality

$$|\phi'(\nu)| \leq R\Delta\phi(\nu) \quad \forall \nu \in [0, 1] \quad (39)$$

The solution to (39) yields the exponential bounds

$$\mu'(z_1)e^{-R\Delta\nu} \leq \phi(\nu) \leq \mu'(z_1)e^{R\Delta\nu} \quad (40)$$

Proof of (37): Using the lower bound in (40), we have

$$\int_0^1 \phi(\nu)d\nu \geq \mu'(z_1) \int_0^1 e^{-R\Delta\nu}d\nu = \mu'(z_1)\frac{1 - e^{-R\Delta}}{R\Delta} \geq \frac{\mu'(z_1)}{1 + R\Delta}$$

where the last inequality uses $1 - e^{-x} \geq x/(1 + x)$ for $x \geq 0$.

Proof of LHS of (38): For the weighted integral, we have

$$\int_0^1 (1 - \nu)\phi(\nu)d\nu \geq \mu'(z_1) \int_0^1 (1 - \nu)e^{-R\Delta\nu}d\nu$$

$$= \mu'(z_1)\left[\frac{1}{R\Delta} - \frac{1 - e^{-R\Delta}}{(R\Delta)^2}\right]$$

$$\geq \frac{\mu'(z_1)}{2 + R\Delta}$$

The final inequality follows from the fact that

$$\frac{1}{x} - \frac{1 - e^{-x}}{x^2} \geq \frac{1}{2 + x} \quad \forall x \geq 0.$$

Proof of RHS of (38): Using the upper bound in (40), we have

$$\int_0^1 (1 - \nu)\phi(\nu)d\nu \leq \mu'(z_1) \int_0^1 (1 - \nu)e^{R\Delta\nu}d\nu$$

$$= \frac{e^{R\Delta} - 1 - R\Delta}{(R\Delta)^2}\mu'(z_1)$$

$$\leq e^{R^2\Delta^2}\mu'(z_1).$$

where we used $e^x - 1 - x \leq x^2 e^{x^2}$ for $x \geq 0$.

The case for $z = z_2$ follows by symmetry. $\quad\square$

**Lemma 9** (Ville [1939])**.** *Let $\{M_t\}_{t=0}^{\infty}$ be a supermartingale with $M_t \geq 0$ almost surely for all $t \geq 0$. Then, for any $\varepsilon > 0$,*

$$\Pr\left[\sup_{t \in \mathbb{N}} M_t \geq \varepsilon\right] \leq \frac{\mathbb{E}[X_0]}{\varepsilon}.$$

**Lemma 10.** *Let $P = \mathcal{N}(\mathbf{0}, \eta H^{-1})$ be a Gaussian distribution with mean $\mathbf{0} \in \mathbb{R}^d$ and covariance $\eta H^{-1}$, where $H \succcurlyeq \lambda I_d$ is a positive definite matrix and $\eta > 0$. Then, if $\lambda \geq 32d\eta c/7$ we have*

$$\mathbb{E}_{\theta \sim P}\left[\exp\left(c\|\theta\|_2^2\right)\right] \leq \frac{4}{3}.$$

*Proof of Lemma 10.* Let $\theta \in \mathbb{R}^d$ be a random variable sampled from $P$. One can verify that it also follows the sample distribution as

$$\sum_{i=1}^{d} \sqrt{\eta \lambda_i} X_i \mathbf{e}_i \quad \text{and} \quad X_i \overset{i.i.d.}{\sim} \mathcal{N}(0,1), \ \forall i \in [d],$$

where $\{\mathbf{e}_i\}_{i=1}^d$ is a set of orthogonal base and $\lambda_i$ is the $i$-th largest eigenvalue of $H^{-1}$. Then, we have

$$\mathbb{E}_{\theta \sim P}\big[\exp\big(c\|\theta\|_2^2\big)\big] = \mathbb{E}_{X_i \sim \mathcal{N}(0,1)}\left[\prod_{i=1}^{d} \exp\big(c\eta \lambda_i X_i^2\big)\right] \leq \left(\mathbb{E}_{X \sim \mathcal{N}(0,1)}\big[\exp\big(c\eta X^2/\lambda\big)\big]\right)^d$$

$$= \left(\mathbb{E}_{Z \sim \chi^2}\left[\exp\left(\frac{c\eta}{\lambda}Z\right)\right]\right)^d \leq \mathbb{E}_{Z \sim \chi^2}\left[\exp\left(\frac{c\eta d}{\lambda}Z\right)\right] \leq \frac{4}{3}$$

where $\chi^2$ denotes the chi-squared distribution with degree of freedom 1. The first inequality is because $\max_{i \in [d]} \lambda_i \leq 1/\lambda$ due to the condition $H \succcurlyeq \lambda I_d$. The last second equality is by the Jensen's inequality since $x^d$ is a convex function with respect to $x$. The last inequality is due to the condition $c\eta d/\lambda \leq 7/32$ and the fact that the moment-generating function of the chi-squared distribution $\mathbb{E}[\exp(tZ)] \leq (1-2t)^{-1/2}$ for $t < 1/2$. $\qquad\square$

**Lemma 11** (Lemma 9 of Faury et al. [2022]). *Let* $\lambda \geq 1$ *and* $\{\mathbf{z}_s\}_{s=1}^{\infty}$ *a sequence in* $\mathbb{R}^d$ *such that* $\|\mathbf{z}_s\|_2 \leq Z$ *for all* $s \in \mathbb{N}$. *For* $t \geq 2$ *define* $V_t := \sum_{s=1}^{t-1} \mathbf{z}_s \mathbf{z}_s^\top + \lambda I_d$. *The following inequality holds*

$$\sum_{t=1}^{T} \|\mathbf{z}_t\|_{V_t}^2 \leq 2d(1+Z^2) \log\left(1 + \frac{TZ^2}{d\lambda}\right).$$

**Lemma 12.** *Let* $P$ *be a probability distribution defined over* $\mathbb{R}^d$ *and* $\Delta$ *be the set of all measurable distributions. For any loss function* $\ell : \mathbb{R}^d \to \mathbb{R}$, *we have*

$$-\frac{1}{\alpha} \log\left(\mathbb{E}_{\theta \sim P}[e^{-\alpha \ell(\theta)}]\right) = \mathbb{E}_{\theta \sim P_*}[\ell(\theta)] + \frac{1}{\alpha} \mathrm{KL}(P_* \| P), \tag{41}$$

*where* $P_* = \arg\min_{P' \in \Delta} \mathbb{E}_{\theta \sim P'}[\ell(\theta)] + \frac{1}{\alpha} \mathrm{KL}(P' \| P)$ *is the optimal solution. Furthermore, for any distribution* $Q$ *defined over* $\mathbb{R}^d$, *we have*

$$-\frac{1}{\alpha} \log\left(\mathbb{E}_{\theta \sim P}[e^{-\alpha \ell(\theta)}]\right) = \mathbb{E}_{\theta \sim Q}[\ell(\theta)] + \frac{1}{\alpha} \mathrm{KL}(Q \| P) - \frac{1}{\alpha} \mathrm{KL}(Q \| P_*). \tag{42}$$

*Proof of Lemma 12.* To prove (41), one can check that the optimal solution to the optimization problem on the right-hand side of (41) is given by

$$P_*(\theta) = \frac{P(\theta)e^{-\alpha \ell(\theta)}}{\int_{\theta \in \mathbb{R}^d} P(\theta)e^{-\alpha \ell(\theta)} \mathrm{d}\theta}.$$

Substituting $P_*$ back into the right-hand side yields the desired equality. Alternatively, (41) can be shown by noting that the mix loss is the convex conjugate of the Kullback–Leibler divergence [Reid et al., 2015]. By the definition of the convex conjugate, the equality holds.

To prove the second part of the lemma, namely (42), let $Z = \int_{\theta \in \mathbb{R}^d} P(\theta)e^{-\alpha \ell(\theta)} \mathrm{d}\theta$. We then have

$$\frac{1}{\alpha} \mathrm{KL}(Q\|P) - \frac{1}{\alpha} \mathrm{KL}(Q\|P_*) - \frac{1}{\alpha} \mathrm{KL}(P_*\|P)$$

$$= \frac{1}{\alpha} \mathbb{E}_{\theta \sim Q}\left[\ln\left(\frac{P_*(\theta)}{P(\theta)}\right)\right] - \frac{1}{\alpha} \mathbb{E}_{\theta \sim P_*}\left[\ln\left(\frac{P_*(\theta)}{P(\theta)}\right)\right]$$

$$= \frac{1}{\alpha} \mathbb{E}_{\theta \sim Q}\left[\ln\left(\frac{e^{-\alpha \ell(\theta)}}{Z}\right)\right] - \frac{1}{\alpha} \mathbb{E}_{\theta \sim P_*}\left[\ln\left(\frac{e^{-\alpha \ell(\theta)}}{Z}\right)\right]$$

$$= \mathbb{E}_{\theta \sim P_*}[\ell(\theta)] - \mathbb{E}_{\theta \sim Q}[\ell(\theta)].$$

where the second equality is due to the fact that $P_*(\theta)/P(\theta) = e^{-\alpha \ell(\theta)}/Z$ for all $\theta \in \mathbb{R}^d$. Then, rearranging the above displayed equation gives us

$$\mathbb{E}_{\theta \sim P_*}[\ell(\theta)] + \frac{1}{\alpha} \mathrm{KL}(P_*\|P) = \mathbb{E}_{\theta \sim Q}[\ell(\theta)] + \frac{1}{\alpha} \mathrm{KL}(Q\|P) - \frac{1}{\alpha} \mathrm{KL}(Q\|P_*),$$

which completes the proof by using (42). $\qquad\square$

**Lemma 13** (Theorem 1.8.2 of Ihara [1993])**.** *The Kullback-Leibler divergence between two $d$-dimensional Gaussian distributions $P = \mathcal{N}(\mathbf{u}_p, \Sigma_P)$ and $Q = \mathcal{N}(\mathbf{u}_q, \Sigma_q)$ is given by*

$$\mathrm{KL}(Q\|P) = \frac{1}{2}\left( \ln\left( \frac{|\Sigma_p|}{|\Sigma_q|} \right) + \mathrm{Tr}(\Sigma_q \Sigma_p^{-1}) + \|\mathbf{u}_p - \mathbf{u}_q\|_{\Sigma_p^{-1}}^2 - d \right).$$

# D   More Discussions on Lee and Oh [2025a]

For MNL bandits, [Lee and Oh, 2025a] (v3 version, the latest one available before the NeurIPS submitted date) claimed an $\mathcal{O}(\sqrt{\log T})$ improvement in the regret bound, which could potentially be applied to logistic bandits to achieve an $\mathcal{O}(\log T \sqrt{T/\kappa_*})$ bound with $\mathcal{O}(1)$ computational cost. However, their analysis relies on a specific upper bound on the normalization factor of a truncated Gaussian, which may not always hold. We elaborate on the main technical issue below in the context of the logistic bandit problem.

With slight abuse of notation, we define the logistic loss as $\ell_s(\theta) = r_t X_t^\top \theta + \log(1 + \exp(X_t^\top \theta))$, where $X_t$ is the action selected by the learner and $r_t \in \{0, 1\}$ is the observed reward. Specifically, building upon the framework of Zhang and Sugiyama [2023], the Lemma C.1 of Lee and Oh [2025a] shows that the estimation error of their estimator satisfies

$$\|\theta_{t+1} - \theta_*\|_{H_{t+1}}^2 \lesssim \underbrace{\sum_{s=1}^t \ell_s(\theta_*) - \sum_{s=1}^t \bar{\ell}_s(\widetilde{z}_s)}_{\texttt{term (a)}} + \underbrace{\sum_{s=1}^t \bar{\ell}_s(\widetilde{z}_s) - \sum_{s=1}^t \ell_s(\theta_{s+1})}_{\texttt{term (b)}},$$

where $\bar{\ell}_s(z) = r_s z + \log(1 + \exp(z))$. In Lee and Oh [2025a], the intermediate term is chosen as $\widetilde{z}_s = \sigma^+\left(\mathbb{E}_{\theta \sim P_s}[\sigma(X_s^\top \theta)]\right)$, where $\sigma(z) = 1/(1 + e^{-z})$ and $\sigma^+(p) = \log(\frac{p}{1-p})$.

A key step of their analysis lies in the choice of $P_s$, which ensures that term (a) and term (b) can be bounded separately. In contrast to the Gaussian distribution used in Zhang and Sugiyama [2023], Lee and Oh [2025a] take $P_s$ as a truncated Gaussian, thereby allowing the bound on term (a) to avoid the additional $\mathcal{O}(\sqrt{T})$ factor incurred in Zhang and Sugiyama [2023]. However, when analyzing term (b), their argument relies on a condition for the normalization factor of the truncated Gaussian distribution, which does not holds in general (see Eqn (C.15) in their paper).

For completeness, we restate it below: there exists a constant $\mathfrak{C}_s \geq 1$ such that

$$\int_{\|\theta\|_{H_s} \leq \frac{3}{2}\gamma} e^{-\frac{1}{2c}\|\theta\|_{H_s}^2}\, \mathrm{d}\theta \leq \int_{\|\theta\|_{H_s} \leq \frac{1}{2}\gamma} e^{-\frac{\mathfrak{C}_s}{2c}\|\theta\|_{H_s}^2}\, \mathrm{d}\theta, \tag{43}$$

where $\gamma, c > 0$ are certain constants and $H_s$ is a symmetric positive-definite matrix. Condition (43) plays an important role in ensuring that the term $\log(Z_s/\widehat{Z}_{s+1})$ in Eq. (C.17) of the paper remains non-positive and does not affect the final bound of term(b). However, it is not evident how to select $\mathfrak{C}_s \geq 1$ to guarantee this property holds throughout, as the left-hand side integrates over a strictly larger region while the integrand decays more slowly. We also mention that this issue was also concurrently addressed by the authors in the later version [Lee and Oh, 2025b] (v5 version, posted at June 2025), and the fixed technique is essentially similar to the mixability-based analysis in our work.

# E   Additional Experimental Details

In this section, we provide additional experimental details and results.

## E.1   Experimental Setup

**Implementation Details.**    All the experiments were conducted on Intel Xeon Gold 6242R processors (40 cores, 4.1GHz base frequency). The algorithms were implemented in Python, utilizing the `scipy` library for numerical computations, such as solving non-linear optimization problems and calculating vector norms, and employing `np.linalg.pinv` to compute the pseudo-inverse of matrices. The running time was measured using the `time` library. The shaded regions in the regret plots represent 99% confidence intervals, computed from 10 independent runs with different random seeds.

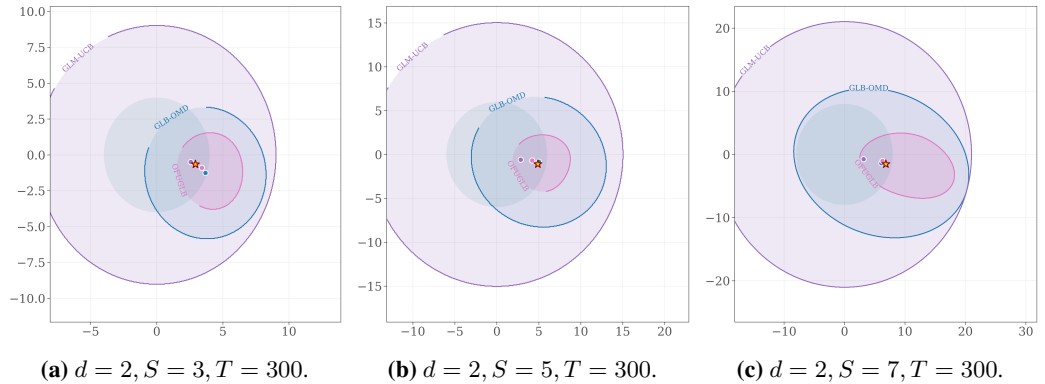

**(a)** $d = 2, S = 3, T = 300$.    **(b)** $d = 2, S = 5, T = 300$.    **(c)** $d = 2, S = 7, T = 300$.

**Figure 3:** Confidence Region of Parameter Estimation.

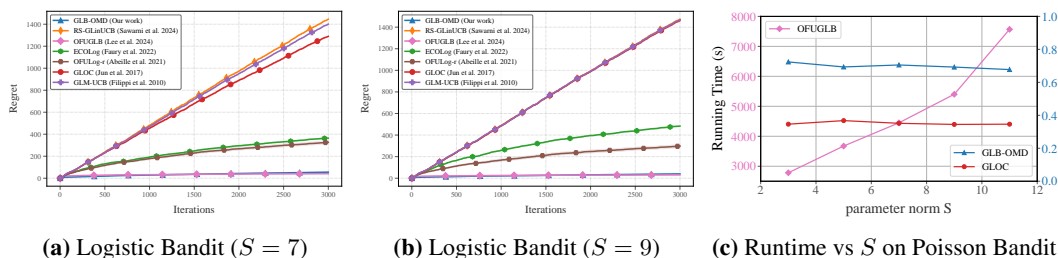

**(a)** Logistic Bandit ($S = 7$)    **(b)** Logistic Bandit ($S = 9$)    **(c)** Runtime vs $S$ on Poisson Bandit

**Figure 4:** Regret and Running Time Dependence on $S$.

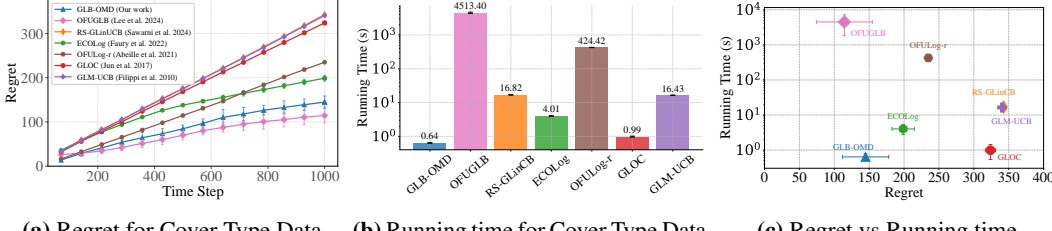

**(a)** Regret for Cover Type Data    **(b)** Running time for Cover Type Data    **(c)** Regret vs Running time

**Figure 5:** Performance comparison of different algorithms on Cover Type Data

**Algorithm Configuration.**    Throughout our experiments, all algorithm parameters were configured according to their theoretical derivations without additional fine-tuning, with the sole exception of the regularization parameter $\lambda$. To ensure a fair comparison, we adopted a unified approach for setting $\lambda$ across different algorithm categories: we set $\lambda = d$ for all efficient online algorithms (including GLB-OMD, RS-GLinCB, ECOLog, and GLOC), while using $\lambda = d \log(1 + t)$ for offline algorithms that require regularization. This distinction reflects the practical consideration that real-world scenarios often exhibit more favorable conditions than the worst-case assumptions.

### E.2    More results on Synthetic Data

To visualize the accuracy of parameter estimation of the algorithms, we plot the confidence region of the parameter estimation for each algorithm in Figure 3. For illustration purposes, we only plot the confidence regions of our algorithm GLB-OMD, the theoretically optimal OFUGLB, and the classical GLM-UCB. We observe that both GLB-OMD and OFUGLB achieve a substantially smaller confidence region than GLM-UCB, indicating that our algorithm achieves an accurate parameter estimation comparable to the statistically optimal.

To further investigate the impact of parameter $S$ on algorithm performance, we conduct additional experiments on logistic bandit tasks with larger $S$ values (Figures 4a and 4b) and analyze the computational time scaling for Poisson bandits (Figure 4c).

The regret curves in Figures 4a and 4b consistently demonstrate that our algorithm maintains its competitive performance regardless of $S$ variations, aligning with the trends observed in our main results. Notably, the regret does not exhibit significant growth as $S$ increases, suggesting the robustness of our approach to parameter scaling. We also note that the performance of RS-GLinCB is very sensitive to the parameter of $S$. This underperformance can be attributed to the fact that the warm-up period of RS-GLinCB is heavily dependent on the constant $S$ and $\kappa$ [Sawarni et al., 2024, Lemma 4.1].

The runtime curves in Figure 4c reveals two key findings: First, our algorithm's running time remains stable (under 1 second for $T = 3000$) across different $S$ values in Poisson bandit tasks. Second, in contrast to this consistent performance, OFUGLB exhibits a pronounced computational overhead that scales with $S$ (requiring 2783 seconds at $S = 3$ compared to 7568 seconds at $S = 9$). This divergence can be attributed to the confidence radius construction in OFUGLB:

$$\mathcal{C}_t(\delta) := \left\{ \theta \in \Theta : \mathcal{L}_t(\theta) - \mathcal{L}_t(\widehat{\theta}_t) \leq \beta_t(\delta)^2 \right\}, \tag{44}$$

where $\beta_t(\delta)^2 = \log \frac{1}{\delta} + \inf_{c_t \in (0,1]} \left\{ d \log \frac{1}{c_t} + 2SL_t c_t \right\} \leq \log \frac{1}{\delta} + d \log \left( e \vee \frac{2eSL_t}{d} \right)$. For Poisson bandits specifically, $L_t = e^S t + \log(d/\delta)$. This results in increasing cost in the optimization steps during the arm selection $X_t = \arg\max_{\mathbf{x} \in \mathcal{X}_t} \max_{\theta \in \mathcal{C}_t(\delta)} \mathbf{x}^\top \theta$, as the algorithm needs to navigate a rapidly expanding nonconvex confidence region.

Overall, our algorithm demonstrates comparable statistical performance to the theoretically optimal OFUGLB while offering substantially improved computational efficiency.

### E.3 Experiment on Forest Cover Type Data

In this experiment, we evaluate our proposed algorithm on the Forest Cover Type dataset from the UCI Machine Learning repository [Blackard, 1998]. This dataset comprises 581,012 labeled observations from different forest regions, with each label indicating the dominant tree species.

Following the preprocessing steps described in Filippi et al. [2010], we centered and standardized the 10 non-categorical features and appended a constant covariate. To enhance the diversity of the arm set and strengthen the experimental results, we partitioned the data into $K = 60$ clusters using unsupervised $K$-means clustering, with the cluster centroids serving as the contexts for each arm. For the logistic reward model, we binarized the rewards by assigning a reward of 1 to data points labeled as the second class ("Lodgepole Pine") and 0 otherwise. The reward for each arm was then computed as the average reward of all data points within its corresponding cluster, yielding reward values ranging from 0.103 to 0.881.

For this task, we set the horizon to $T = 1000$ and the confidence parameter to $\delta = 0.01$. After analyzing the data, we set $S = 6$ and $\kappa = 200$. We evaluated our algorithm against the same baselines used in the logistic bandit simulation experiment, running each method over 10 independent trials and averaging the results to report the regret and the running time. The error bars in the figures denote 99% confidence intervals for both regret and runtime.

Compared to synthetic data, real-world datasets exhibit higher noise and complexity, demanding careful exploration-exploitation trade-offs. Thus, we shrank the estimated confidence set of all the algorithms in a comparable way to achieve a better balance between exploration and exploitation in this real-world dataset. Traditional GLB algorithms are particularly sensitive to noise, often leading to excessive exploration and higher regret.

Figure 5a presents the regret progression of different algorithms over time, while Figure 5b compares their computational efficiency. Figure 5c further illustrates the regret-time trade-off for our method. The results demonstrate that our algorithm achieves significantly faster runtime without compromising robustness or performance, even in noisy environments.

