# OpenReview forum: "Generalized Linear Bandits: Almost Optimal Regret with One-Pass Update"
_NeurIPS.cc/2025/Conference — NeurIPS 2025 poster_

### Official Review · Reviewer_fAEe · 2025-06-26

**Clarity:** 3
**Significance:** 2
**Originality:** 2
**Rating:** 4
**Confidence:** 3

**Summary:**

In this paper, the authors study the GLB problem and propose the first algorithm with $O(1)$ per-round computational complexity, achieved by leveraging the OMD estimator.
The main technical contribution is an improvement of the regret bound by a factor of $\sqrt{\log t}$ over the prior result of Zhang and Sugiyama [2023].

**Questions:**

Please see the questions listed in the Weaknesses section above.

**Ethical Concerns:**

["NO or VERY MINOR ethics concerns only"]

**Final Justification:**

Most of my concerns have been successfully addressed, particularly with respect to the technical contribution. I still have some questions about the significance of the $\sqrt{\log t}$ improvement; however, as it stands, I believe the technical contribution is reasonable to merit acceptance.

Although the lack of code is somewhat disappointing, I understand that sharing it may not be feasible under this year’s NeurIPS policies. Therefore, I will maintain my positive score.

**Limitations:**

Yes.

**Paper Formatting Concerns:**

No issues.

**Quality:**

2

**Strengths And Weaknesses:**

**Strengths**

This paper is overall well-organized and easy to follow. The main technical results are supported by adequate claims, and the empirical results further strengthen the theoretical guarantees.


**Weaknesses**

- The main technical contribution seems to be the improvement by a $\sqrt{\log t}$ factor compared to Zhang and Sugiyama [2023].
Are there any additional technical challenges in adapting the OMD estimator from Zhang and Sugiyama [2023] to the GLM setting?
Since the overall algorithm and analysis closely follow those in Zhang and Sugiyama [2023], I am not fully convinced that the $\sqrt{\log t}$ improvement alone constitutes a substantial contribution.

- No code is provided.

- In the experiments, why wasn’t the proposed algorithm compared with that of Zhang and Sugiyama [2023]?

     In Figure 1(a) and 1(b), why does RS-GLinUCB underperform in the case of $S=3$ compared to $S=5$?

     Although the values of $\kappa$ are smaller in the Poisson bandit setting than in the logistic case, warm-up-based methods like RS-GLinUCB completely fail.  Why does this happen?

---

> ### Author Rebuttal · Authors · 2025-07-31
>
> Thank you for the helpful comments! We will highlight the technical contributions in comparison with Zhang and Sugiyama, and provide additional details on the experimental comparisons. The detailed responses are provided below:
>
> ---
>
> **Q1**: Are there any additional technical challenges in adapting the OMD estimator from Zhang and Sugiyama [2023] to the GLM setting?
>
> **A1:** Thank you for the comments. We would like to take this opportunity to further highlight the main technical contributions of our paper. We note that the analysis in Zhang and Sugiyama [2023] **is not applicable** to the GLB setting, even with the addition of an $O(\sqrt{\log t})$ term. This is because their construction of the intermediate term relies heavily on the specific structure of the logistic loss and builds upon results from online logistic regression [Foster et al., 2018]. In the GLB setting, however, it remains unclear whether a good online estimator exists.
>
> One of our key contributions is the introduction of the **mix loss** (Equation (9)) as an intermediate term. This allows us not only to extend the analysis to the more general GLB setting, but also, as a byproduct, to improve the $O(\sqrt{\log T})$ factor in the logistic bandit setting compared to the result in Zhang and Sugiyama [2023]. When handling the decomposition based on the mix loss, we employ a different strategy: we analyze term (a) using Ville’s inequality, and we provide a simplified analysis for term (b), which avoids the lengthy calculations appearing in Zhang and Sugiyama’s analysis.
>
> ---
>
> **Q2**: No code is provided.
>
> **A2:** Thank you for the interest! We will release the code along with the next version of the paper.
>
> ---
>
> **Q3**: In the experiments, why wasn’t the proposed algorithm compared with that of Zhang and Sugiyama \[2023]?
>
> **A3:** When applied to logistic bandit tasks, the primary difference between the two algorithms lies in the confidence set radius. Our proposed GLB-OMD algorithm achieves a tighter confidence bound of $O(\log t)$, compared to the $O((\log t)^{3/2})$ bound in Zhang and Sugiyama \[2023]. Theoretically, $\beta_t^{\text{GLB-OMD}}(\delta) < \beta_t^{\text{MLogB}}(\delta)$, which ensures that the performance of GLB-OMD will be better. That is also why we exclude OFULog+ (Lee et al. 2023) from the benchmark. (OFULog+ is similarly constructed as OFUGLB and is theoretically worse than OFUGLB)
>
> To highlight this, we conducted additional experiments comparing the regret performance of GLB-OMD (our method) and OFUL-MLogB (Zhang and Sugiyama, 2023). Furthermore, after the initial submission, we found that the step size $\eta$ for GLB-OMD can be refined to $\eta = 1 + RS$, replacing the original value $\eta = 2(1 + RS)$ used in Theorem 1. This refinement results in a tighter confidence set and improves the regret bound by a constant factor. We also included the refined results in the comparison. Due to time constraints, these experiments were run for 3000 rounds across 10 trials and included only computationally efficient algorithms. The final-round regret of these efficient algorithms is reported below.
>
> | S    | GLB-OMD (Our work) | OFUL-MLogB (Zhang and Sugiyama, 2023) | ECOLog (Faury et al. 2022) | GLOC (Jun et al. 2017) | GLB-OMD (Refined $\eta$) |
> | ---- | ------------------ | ------------------------------------- | -------------------------- | ---------------------- | ------------------------ |
> | 3    | 227.42             | 236.04                                | 229.30                     | 347.07                 | 105.78                   |
> | 5    | 146.44             | 169.84                                | 224.28                     | 708.92                 | 82.58                    |
> | 7    | 99.03              | 114.73                                | 201.06                     | 1291.06                | 56.67                    |
> | 9    | 81.60              | 92.82                                 | 191.68                     | 1459.46                | 40.06                    |
>
> ---
>
> **Q4**: In Figure 1(a) and 1(b), why does RS-GLinUCB underperform when $S = 3$ compared to $S = 5$?
>
> **A4:** Thank you for the questions. The RS-GLinUCB method incorporates a warm-up procedure (Algorithm 2, lines 4–8). In the experiments, the warm-up condition is typically triggered during the early stages of the learning process. The number of warm-up iterations is on the order of $O(\kappa S^3 (\log T)^2)$ as shown in Sawani et al., 2024. As a result, when $S = 5$, the method performs worse than when $S = 3$ during the initial phase, since more iterations are spent on warm-up. However, in the later stage, after the warm-up is mostly complete, the regret of the method follows the theoretical bound of $O(\sqrt{T/\kappa_*})$. Given that $\kappa_* \propto e^S$, the RS-GLinUCB method with $S = 5$ will ultimately outperforms the case with $S = 3$ when $T$ is large enough.
>
> ---
>
> **Q5**: Although the nonlinear values are smaller in the Poisson bandit setting than in the logistic case, warm-up-based methods like RS-GLinUCB completely fail. Why does this happen?
>
> **A5:** The theoretical guarantee of RS-GLinUCB holds only when the rewards are almost surely bounded by a constant $R_{\mathrm{reward}}$ , which does not apply in the Poisson bandit setting as the reward can be unbounded. To empirically evaluate the method in our experiments, we set $R_{\mathrm{reward}}$ to ensure that the randome rewards are bounded with high probability within $T = 3000$ rounds, and used this value in the algorithm’s parameter configuration.
>
> However, as shown in Theorem 4.2 of the RS-GLinUCB paper, the regret scales as $O(R_{\mathrm{reward}} ^5)$. This high dependence on $R_{\mathrm{reward}}$  leads to poor performance when the algorithm is applied to the unbounded Poisson bandit setting.

---

> > ### Comment · Reviewer_fAEe · 2025-08-04
> >
> > Thank you for the detailed response. Most of my concerns have been addressed, so I will maintain my positive score.

---

> > > ### Author Response · Authors · 2025-08-05
> > >
> > > Thank you again for the insightful review and positive evaluation of our paper! We will highlight our contributions and include additional explanation of the experiments in the revision.

---

### Official Review · Reviewer_JoLY · 2025-07-01

**Clarity:** 3
**Significance:** 3
**Originality:** 2
**Rating:** 5
**Confidence:** 2

**Summary:**

The authors propose a new algorithm for the General Linearized Bandits setting which enjoys a balance of both computaional and statistical efficiency improvements over previous methods.

Specifically, they propose an OMD-based estimator for a confidence set for the latent parameter $\theta_*$ which provides tighter confidence sets than previous approaches by leveraging self-concordance. They use this estimator to provide uncertainty sets for a UCB-style algorithm that, together with the estimator, has constant memory and time costs while achieving competitive regret performance with methods requiring orders of magnitude more compute.

They provide theoretical analysis and experiemntal results on synthetic tasks to demonstrate the gains assocaited with their approach.

**Questions:**

My primary concerns with this paper are in its writing clarity and organisation. Addressing the weaknesses outlined above are sufficient grounds for me to increase my clarity score.

**Ethical Concerns:**

["NO or VERY MINOR ethics concerns only"]

**Final Justification:**

This paper generalizes OMD + UCB to the broader GLB setting. They improve SotA for logistic bandits across the regret, time, memory frontier, and include regret analysis with impressive empirical results across both statistical and computational fronts. During the rebuttal period, the authors engaged with my questions and provided thoughtful feedback that improved the paper and addressed my concerns. For these reasons, I recommend acceptance.

**Limitations:**

Yes

**Paper Formatting Concerns:**

No formatting concerns.

**Quality:**

3

**Strengths And Weaknesses:**

*Strengths:*
- Generalizes OMD + UCB approach to the broader GLB setting
- Improves SotA for logistic bandits across the regret, time, memory frontier
- Regret analysis with impressive empirical results across both statsitical and computational fronts (1000x test time computational improvement)
- Leveraging self-concordance for inproved confidence sets is elegant and general

*Weaknesses*

- Although the core contributions of this paper are strong, overall the paper reads as somewhat rushed in its organsation and preparation. For example, the introduction was harder to read than necessary, with many of the core definitions deferred to the preliminaries in a way that made it hard to appreciate what exactly was being estimated and how that estimate was being used within a UCB framework. This feels unneccessary since it ultimately is quite a simple quantity to describe in one or two sentences. Generally, assumptions in the first few pages of the paper felt scattered in the prose. For example, it wasn’t clear to me if the link function $\mu$ was known to the agent, belonged to a family etc. until the preliminaries. I found it hard to apprecaite what the setting was exactly until multiple passes. In short, the full setting only became clear to me upon reading the preliminaries; the introduction felt confusing.
- Some technical assumptions are made in the notation which never feature in the prose of the paper, e.g. the action sets $\mathcal{X}_t$ appear to be non-stationary given the $t$ subscripts, yet there is no motivaiton for this consideration and no regard to it in the description of the setting. This felt disracting.
- The paper contains some instances of vague expersions such as “non-linearity is beneficial” which made for a somewhat frustrating read.
- Phrases like “bridging the loss gap” are unclear until deeper into the proof section and made for a confusing read.
- Some sentences seem incomplete, e.g. “We also note that OMD-based online estimators have been used to develop jointly efficient algorithms in the logistic bandit setting … their analyses of the confidence set rely heavily on the specific structure of the logistic link function, limiting their applicability to more general GLB models.” lacks a bridge word such as “but” or “however”
- Figures lack clarity, the shaded region should be well defined.
- There are some notation inconsistencies, such as starting wth $\mathbf{x}$ early in the paper for actions but then switching to $X_t$ later, and mentioning that the notation is due to actions being “random variables”, which didn’t help me apprecaite why earlier notation was different; also $\tau$ is not defined anywhere in the paper yet appears in important definitions.
- They say “This bound is nearly optimal in terms of the horizon $T$ up to logarithmic factors” but give no specific citation for this claim or substantiate it.

---

> ### Author Rebuttal · Authors · 2025-07-31
>
> Thank you for appreciating the technical contributions of our work and for the helpful comments! The main concerns raised relate to the clarity of (1) the problem setup and (2) the explanation of UCB in the introduction. In the revision, we will improve the introduction and refine several vague expressions throughout the paper. Below, we outline our detailed revision plans.
>
> ---
>
> **Q1: Regarding the writing in the introduction section**
>
> > the core definitions deferred to the preliminaries ... what exactly was being estimated and how that estimate was being used within a UCB framework
>
> > if the link function was known to the agent, belonged to a family etc.
>
> **A1:** Thank you for the constructive feedback. We would revise the introduction to make the problem setup and issue of parameter estimation clearer. We plan to make the following revision:
>
> - Add more details about the problem setup and assumptions at the first paragraph
>
> $\underline{\mbox{[Revision for first paragraph]}}$: ...receiving feedback in the form of rewards. In this paper, we study the contextual multi-armed bandit problem under the framework of generalized linear models (GLMs). In this setting, each action is characterized by a contextual feature vector $\mathbf{x} \in \mathcal{X}\_t \subset \mathbb{R}^d$, **where the arm set $\mathcal{X}_t$ can vary over time**. The learning process can be seen as a $T$ round game between the learner and the environment. For each round $t$, upon selecting an action $X_t\in\mathcal{X}\_t$ , the learner receives a stochastic reward $r_t \in \mathbb{R}$, which is generated according to a GLM (see definition in Section 2.1).* The goal of the learner is to maximize the cumulative expected reward obtained over the time horizon $T$. Under the GLM model, the expectation of the reward satisfies $\mathbb{E}[r_t\,|\,X_t] = \mu(X_t^\top \theta_*)$, where $\mu: \mathbb{R} \to \mathbb{R}$ is a non-linear link determined by the GLM model and **is known to the learner**. The unknown part is the underlying parameter $\theta_* \in \mathbb{R}^d$ , **which needs to be estimated by the learner from the observed action-reward pairs.**...
>
> [*footnote:*] $*$: To highlight its stochastic nature, we denote the action selected by the learner as $X_t$, in contrast to the deterministic case, for which we use $\mathbf{x}$.
>
> - Add more details about the UCB method at the second paragraph
>
> $\underline{\mbox{[Revision for second paragraph]}}$: A canonical solution to the Generalized Linear Bandit (GLB) problem is the GLM-UCB algorithm [Filippi et al., 2010], which belongs to the family of UCB-type methods [Agrawal, 1995; Auer, 2002]. Specifically, at each iteration $t \in [T]$, the algorithm **first estimates the true parameter $\theta_*$** **using maximum likelihood estimation based on the historical data** $\\{(X_s, r_s)\\}\_{s=1}^{t-1}$ and yields an estimator $\theta_t$. Using $\theta_t$, it then computes the empirical expected reward for each arm and constructs an upper confidence bound (UCB) to guide arm selection. The GLM-UCB algorithm achieves a regret bound of $\mathcal{O}(\kappa \sqrt{T})$, where the constant $\kappa = 1/\inf_{\mathbf{x}\in\cup_{t=1}^T\mathcal{X}\_t,\theta\in\Theta} \mu'(\mathbf{x}^\top\theta)$ captures the non-linearity of the link function $\mu$. While GLM-UCB is nearly optimal in its dependence on $T$, the non-linearity of the link function raises significant concerns regarding both computational and statistical efficiency...
>
>
> ---
>
> **Q2: About the time-varying arm set $\mathcal{X}_t$.**
>
> **A2:** Thank you for pointing this out. We use $\mathcal{X}_t$ to indicate that we consider a setting where the arm set may change over time. A time-varying arm set is arguably one of the most general settings in the linear bandits literature, as it does not require any distributional assumptions on the action set. This setting aligns with many works on (generalized) linear bandits, such as [1,2,3]. In the revised version of the paper, we will make this dynamic action set assumption more explicit and clearly motivate it in the main text.
>
> $\underline{\mbox{[Revision at Section 2.1]}}$: ...A stochastic reward $r_t \in \mathbb{R}$. Here, we use the notation $\mathcal{X}_t$ to indicate that the arm set can **change over time**. This modeling assumption allows us to capture many common scenarios in real-world applications. For example, in product recommendation systems, the items may be dynamically added or removed over time. This requires the algorithm to adapt to the current set of available options.
>
> [1] Improved algorithms for linear stochastic bandits. Abbasi-Yadkori et al., 2011
>
> [2] Improved optimistic algorithms for logistic bandits. Faury et al., 2020.
>
> [3] A Unified Confidence Sequence for Generalized Linear Models, with Applications to Bandits. Lee et al., 2024.
>
> ---
>
> **Q3: some instances of vague expersions**
>
> Thank you for highlighting this. We will made the following revision:
>
> > line 132: ".. $\kappa_*$ appears at the denominator, indicating that the non-linearity is beneficial to regret minimization"
>
> -->" $\kappa_*$ appears at the denominator, **which largely improves the $O(\kappa\sqrt{T})$ bound by Filippi et al., [2010]**"
>
> > Line 304 "In our analysis this term is similarly instrumental in bridging the cumulative loss gap between..., thereby enabling a sharp bound for the estimation error."
>
> -->In our analysis, the mix loss is instrumental in analyzing the *inverse regret* defined in (9). In particular, the decomposition based on the mix loss $m_s(P_s)$ enables a more general and analytically versatile formulation than $\ell_s(\tilde{\theta}_s)$ used in (9). The former reduces to the latter when $P_s$ is chosen as a Dirac distribution, while more carefully chosen distributions can lead to tighter bounds.
>
> ---
>
> **Q4: Some sentences seem incomplete**
>
> **A4:** We will make the revision as follows:
>
> > line 58-60: " ... with Bernoulli rewards, their analyses of the confidence set rely heavily on the specific structure of the logistic link function $\mu(z) = 1/(1+e^{-z})$, limiting their applicability to more general GLB models."
>
> $\underline{\mbox{[Revision for Line 58-62]}}:$ We also note ..., a special case of GLB with Bernoulli rewards. **However,** their analyses of the confidence set rely heavily on the specific structure of the logistic link function $\mu(z) = 1/(1+e^{-z})$, which limits their applicability to more general GLB models.
>
> ---
>
> **Q5: Figures lack clarity, the shaded region should be well defined.**
>
> **A5:** We appreciate this suggestion. In the revised version, we will add detailed figure captions. Specifically, the shaded regions in the regret plots represent the 99% confidence intervals, based on 10 independent runs with different random seeds. In the Covertype Regret-vs-Time figure, the error bars represent the 99% confidence intervals for both regret and runtime; we visually scaled the error bars (×2 for regret, ×6 for runtime) to enhance visibility. This will be clearly stated in the figure descriptions.
>
> ---
>
> **Q6**: Notations about $\mathbf{x}$ and defiition of $\tau$
>
> **A6:** We will add the definition of $\tau$ as below:
>
> $\underline{\mbox{[Revision at Line 95-96 for the defnition of $\tau$ ]:}}$ ...$g:\mathbb{R}\rightarrow\mathbb{R}$ is the dispersion function controlling the variability of the distribution **and $\tau\in\mathbb{R}$ is a parameter that governs the variability of the distribution**.
>
> As for the notation of the action, we follow the convention used by Abbasi-Yadkori et al. (2011), where $X_t$ denotes the learner's action as it is a random variable, while $\mathbf{x}$ represents a deterministic action. To prevent any confusion, we will include a footnote the first time $X_t$ appears to clarify this distinction.
>
> **Ref:** Improved Algorithms for Linear Stochastic Bandits. Abbasi-Yadkori et al., 2011.
>
> ---
>
> **Q7**: “This bound is nearly optimal in terms of the horizon up to logarithmic factors” but give no specific citation for this claim or substantiate it.
>
> **A7:** Thank you for pointing this out. For the logistic bandits problem, Abeille et al. (2022) established a lower bound of $\Theta(\sqrt{T/\kappa_*})$. Since logistic bandits are a special case of GLBs, this implies that the lower bound for GLBs is at least $\Theta(\sqrt{T/\kappa_*})$. In this sense, the $O(\kappa \sqrt{T})$ upper bound by Filippi et al. (2010) is nearly optimal in terms of the dependence on $T$, but suboptimal with respect to the dependence on $\kappa$. We will clarify this point in the next version.
>
> **Ref:** Instance-Wise Minimax-Optimal Algorithms for Logistic Bandits. Abeille et al., 2022.

---

> ### Comment · Reviewer_JoLY · 2025-08-06
>
> Thank you for addressing all of my questions and concerns. I will raise my score and recommend acceptance.

---

> > ### Author Response · Authors · 2025-08-07
> >
> > Thank you for the helpful comments and for raising the score! We will clarify the problem setting more clearly in the introduction.

---

### Official Review · Reviewer_h1Sp · 2025-07-02

**Clarity:** 4
**Significance:** 3
**Originality:** 3
**Rating:** 5
**Confidence:** 4

**Summary:**

The paper introduces a new, tight concentration bound for the online mirror descent (OMD)-based online estimator for generalized linear bandit (GLB) that takes $\mathcal{O}(1)$ space/time complexity per round. The analysis involves appropriately using the notion of mix losses from online learning, which allows it to generalized to GLBs beyond prior works that were only applicable to logistic losses. The new concentration-based UCB algorithm is shown to achieve nearly tight (up to norm factors) regret bounds, and is validated empirically across logistic and Poisson bandits.

**Questions:**

1. How was the runtime computed? Did you use time or timeit?
2. Can the authors plot the confidence regions as well? Also, in line 710-711, the authors state "as the algorithm must navigate a rapidly expanding confidence region." Is it really expanding, though? Because at the end, $\beta_t(\delta) = \Theta(\log t)$ and the number of terms for the log-likelihood loss on the LHS increases linearly.
3. If the arm-set is fixed, then could one hope for slightly better guarantee by any chance, e.g., [1]?


[1] https://arxiv.org/abs/2202.02407

**Ethical Concerns:**

["NO or VERY MINOR ethics concerns only"]

**Final Justification:**

Much of my concerns have been addressed. I maintain my score of 5.

**Limitations:**

yes

**Quality:**

4

**Strengths And Weaknesses:**

**Strengths**
- Well-written
- Technically sound, and the proposed approach to building new concentration seems distinct from prior works.


**Weaknesses**
- Some of the lemmas are lacking the original references. For instance, Lemma 8 is originally Lemma 7 & 8 of Abeille et al. (2021), which in turn was originally proved in Lemma 9 of Faury et al., (2020). Also, the maximal inequality is actually originally known as Ville's inequality due to Ville (1939). Also, the whole idea of utilizing self-concordance is originally due to Bach (2010).
- At the end, the proof proceeds similarly to "regret-to-confidence-set" conversion-type results [1,2] as well as the recent PAC-Bayes argument for GLBs [3]. Especially the overall proof flow of Theorem 1 (martingale, Ville's inequality, Donsker-Varadhan-type characterization of KL, use of KL closed-form) resembles [3], although there are indeed key differences, such as the choice of posterior ([3] chooses uniform while this paper chooses Gaussian? please correct me if I'm way off here). Anyhow, some discussions to such relevant literature would be nice.
- Over the 10 independent trials, some sort of confidence/error interval would be nice addition on top of the average regret! Since 10 is a bit small, maybe bootstrapped confidence interval?
- Later, I hope that the authors would release the codes :)

**Typos**
- Line 629: o $\Rightarrow$ To


[1] https://proceedings.mlr.press/v238/lee24d.html

[2] https://arxiv.org/abs/2504.16555

[3] https://openreview.net/forum?id=MDdOQayWTA

---

> ### Author Rebuttal · Authors · 2025-07-31
>
> Thank you for the appreciation of our work and for the constructive feedback! In the revision, we will add the original reference for the technical lemma and clarify the connection and differences with the previous papers on GLB. Below, we provide detailed responses to your questions.
>
> ---
>
> **Q1** Some of the lemmas are lacking the original references.
>
> **A1:** Thank you for pointing this out. We will correct the citation in the future version. Specifically, we will make the following revision:
>
> - Lemma 8 --> Lemma 8 (Lemma 9 of Faury et al. [2020]);
> - Lemma 9 (Theorem 3.9 of Lattimore and Szepesv&aacute;ri [2020]). --> Lemma 9 (Ville [1939]).
> - Add: Bach [2010] first utilized self-concordance to analyze logistic regression.
>
> ---
>
> **Q2:**  The proof proceeds similarly to ‘regret‑to‑confidence‑set’ conversion‑type results [1, 2] as well as the recent PAC‑Bayes argument for GLBs [3]
>
> **A2:** Thank you for the insightful comments. Indeed, our analysis follows a similar structure to the “regret-to-confidence-set” framework, but with several key distinctions. One major difference is that in [1,2,3], the confidence set is constructed **for the MLE estimator**, whereas our approach builds it for the **OMD estimator.** This fundamental distinction leads to a different construction of the virtual algorithm (or intermediate term) used in the decomposition. In the following, we provide a brief discussion of the differences compared to the work on GLB [2,3].
>
> - Comparison with [3]: Specifically, when comparing with [3], Lemma 3.1 in [3] upper bounds the cumulative loss $\mathcal{L}\_t(\theta_*) := \sum\_{s=1}^t \ell_s(\theta_*)$ using the expected cumulative loss under a fixed uniform distribution $Q$, i.e., $\mathbb{E}\_{\theta \sim Q}[\sum_{s=1}^t \ell_s(\theta)]$. In contrast, we upper bound $\mathcal{L}\_t(\theta_*)$ via the mix loss constructed from a time-varying Gaussian distribution $P_s$. This use of the mix loss and the dynamic nature of $P_s$ is essential in tracking the time-varying OMD estimator throughout our analysis.
>
> - Comparison with [2]: During the rebuttal, we also became aware of a very recent paper [2] (posted on arXiv in April) that similarly uses a mixed loss for confidence set construction. The main difference lies in the design of the distribution $P_s$ for the mix loss: [2] builds $P_s$ based on the exponential weights method to track the MLE estimator, while our distribution $P_s$ is specifically tailored to follow the OMD estimator.
>
> In the next version, we will include a more detailed technical comparison with related work, highlighting both the similarities and the key differences between our approach and prior or concurrent studies.
>
> ---
>
> **Q3:** some sort of confidence/error interval would be nice addition on top of the average regret
>
> **A3**: In Figures (a), (b), (c), and (d), we use **shaded regions** to represent the error intervals.  In the Poisson bandit experiments (Figures (c) and (d)), the error bars are relatively small compared to the scale of the regret, making them nearly invisible. For the Covertype dataset experiments in the appendix, error bars are also included to illustrate variability.
>
> In the revised version, we will include a more detailed explanation of the error bar configuration. Specifically, the shaded regions in the regret plots represent 99% confidence intervals, computed from 10 independent runs with different random seeds. In the Covertype Regret-vs-Time figure, the error bars also denote 99% confidence intervals for both regret and runtime.
>
> ---
>
> **Q4:** Later, I hope that the authors would release the codes. How was the runtime computed? Did you use time or timeit?
>
> **A4:** Thank you for the interest! We will release the code along with the next version. We use "time" to record the running time. We will clarify this in the paper.
>
> ---
>
> **Q5:**  Can the authors plot the confidence regions as well? Also "as the algorithm must navigate a rapidly expanding confidence region." “Is it really expanding, though?
>
> **A5:** Thank you for your suggestions. In the original sentence, our intention was to convey that the confidence set expands **with increasing values of $S$**, rather than over time $t$. As shown in Figure 3(c) for Poisson Bandits, we observe that the running time of OFU-GLB increases with $S$. We attribute this to the fact that the confidence set (Equation 44) used by OFU-GLB grows with $S$, which in turn increases the computational cost of constructing the UCB for each arm. In contrast, the running times of OMD-GLB and GLOC are not affected by $S$, as they employ ellipsoidal confidence sets that allow for closed-form computation of the UCB.
>
> In the next version, we will revise the sentence to: *"...as the algorithm will navigate an expanding confidence region with respect to $S$"*, and we will also include visualizations of the confidence regions for the different methods.
>
> -------
>
> **Q6**: If the arm‑set is fixed, then could one hope for slightly better guarantee by any chance, e.g., [1]?
>
> **A6**: Thank you for the insightful questions. One of the main challenges in obtaining a bound with $O(\sqrt{d})$ dependence on dimensionality appears to be the need for an OMD-based analogue of Lemma 1 from Mason et al., 2022, which itself is essentially derived from Theorem 1 in Jun et al., 2022. If a warm-up procedure is allowed, it may be possible to adapt the proof of [Theorem 1, Jun et al., 2022] to accommodate the OMD estimator. This is a promising and interesting direction for future work, and we will incorporate a discussion of it in the next version.
>
> **Ref:** Improved Confidence Bounds for the Linear Logistic Model and Applications to Bandits. Jun et al., ICML 2022.

---

> ### Comment · Reviewer_h1Sp · 2025-08-04
>
> Thank you for your responses. Most of my concerns have been addressed, and I maintain my score.
>
> Some minor stuff:
> - **A2 Comparison with [3]:** Although the proof starts with Lemma 3.1, after the Donsker-Varadhan, [3] also bounds $\mathcal{L}_t(\theta_*)$ with time-varying Uniform distribution (see Eqn. (7)). It would be good if this can be clarified when the authors put in this comparison.
> - Jun et al. paper is at ICML 2021.
> - [Minor] I believe that the notion of mixability first came from [1].
>
> [1] https://www.sciencedirect.com/science/article/pii/S0022000097915567

---

> > ### Author Response · Authors · 2025-08-05
> >
> > Thank you again for the expert review and helping us better understand the literature! We will clarify that [3] also uses a time-varying distribution $P_t$ in the revision. One distinction is that in [3], the distribution $P_t$ is shared across all individual losses $\\{\ell_s\\}\_{s=1}^t$, as it is used to compute $\mathbb{E}\_{P_t}[\mathcal{L}\_t(\theta)]$ for tracking $\sum_{s=1}^t \ell_s(\theta_t^{\mathsf{mle}})$. In our case, the distributions vary across individual functions, with $\sum_{s=1}^t m_s(P_s)$ designed to track $\sum_{s=1}^t \ell_s(\theta_s)$. We will incorporate your suggestion and provide a careful discussion of related work in the revision.

---

### Official Review · Reviewer_7rDn · 2025-07-06

**Clarity:** 3
**Significance:** 2
**Originality:** 3
**Rating:** 5
**Confidence:** 3

**Summary:**

The paper studies the problem of generalized linear bandits, and proposes an algorithm which is jointly sample-efficient and computationally efficient. The proposed method relies on a new confidence set construction for generalized linear models, which is the main technical contribution of the paper. The new sequence of confidence sets is a sequence of ellipsoids, where each ellipsoid is centered at the estimate returned by online mirror descent (OMD), as opposed to the usual (regularized) maximum likelihood estimator (MLE). The proposed GLB-OMD method is evaluated in several experiments. In these experiments, it is found that the run time of GLB-OMB is a lot lower than that of the state-of-the-art OFUGLB method, yet the regret of GLB-OMD is not much worse than that of OFUGLB.

**Questions:**

By modifying the construction in this paper, would it be possible to construct even smaller confidence sets centered at OMD estimates which are not ellipsoids (perhaps by using a different regularizer/penalty in the OMD update)? For GLMs, restricting the confidence set to be an ellipsoid generally results in looser confidence sets.

The general technique of substituting the log-likelihood loss for the mix-loss to derive confidence sets has appeared in a couple of very recent works (Kirscher et al. 2025, Clerico et al. 2025). I don’t consider this to be a weakness, since these papers were released in close proximity to the submission deadline. Nevertheless, one could add a comment on this to the final version of the paper.

Kirschner, Johannes, et al. "Confidence Estimation via Sequential Likelihood Mixing." arXiv preprint arXiv:2502.14689 (2025).

Clerico, Eugenio, et al. "Confidence Sequences for Generalized Linear Models via Regret Analysis." arXiv preprint arXiv:2504.16555 (2025).

**Ethical Concerns:**

["NO or VERY MINOR ethics concerns only"]

**Final Justification:**

The authors’ rebuttal has addressed the concerns raised in my review. I believe that the paper should be accepted, so I have raised my score.

**Limitations:**

I understand that maximizing the upper confidence bound described in Section 3.1 is in general NP-hard. Indeed, in the linear case, the UCB is a convex function of the action, and convex maximization is in general NP-hard. Given that one of the main claims of the paper is about computational efficiency, it is a bit odd that this is never mentioned.

**Paper Formatting Concerns:**

None.

**Quality:**

3

**Strengths And Weaknesses:**

**Strengths:**
The paper is generally well-written and the results are presented and explained well. In particular, I think that (for the most part) Section 4 does good job of explaining how the construction of the new confidence sequence works and how it differs to previous confidence sequence constructions.

The GLB-OMD method appears to be an effective solution to the problem described in the introduction. Namely, providing a generalized linear bandit algorithm with near-optimal regret and constant (w.r.t. $t$) time and space complexity per round.

**Weaknesses:**
My biggest concern is with the significance of the problem being solved. I’m not sure that replacing a time complexity per round of order $t$ (or $\log t$) with constant (w.r.t. $t$) time complexity is of great important when the time complexity of running the algorithm is largely determined by the cost of maximizing the UCB, unless the action set is finite and small. Moreover, as far as I can tell, the computational challenge of maximizing the UCB is not mentioned anywhere.

I think that some of the claims made are not fully supported or slightly misleading. For instance, Remark 1 fails to mention that, for instance, the OFUGLB method also accommodates unbounded GLMs such as Gaussian or Poisson. On line 221, it is stated that “self-concordant GLBs are no more difficult than linear bandits”. This is true if $T$ is large enough to make the constant term in the regret bound negligible, but this constant term may be extremely large.

Some relevant prior work is not cited. In particular, Proposition 2 is not a new result. For instance, in the book by Grünwald (2007), this result appears under the name “no-hypercompression inequality”.

Grünwald, Peter D. The minimum description length principle. MIT press, 2007.

---

> ### Author Rebuttal · Authors · 2025-07-31
>
> We thank the reviewer for the insightful comments and for bringing the concurrent work to our attention. In the revision, we will include these works for discussion. Below, we address your concerns regarding the significance of the one-pass update and clarify several statements in the paper.
>
> ---
>
> **Q1:** My biggest concern is with the significance of the problem being solved...the computational challenge of maximizing the UCB is not mentioned anywhere.
>
> **A1:** We understand that the computational cost of UCB-type algorithms stems from two components: parameter estimation and arm selection. We agree with the reviewers that identifying the arm that maximizes UCB is generally computationally challenging. Our main contribution is the development of **a one-pass estimator (and confidecne set) for the GLB problem**, which we believe is of both theoretical interest and practical value.
>
> In the revision, we will (1) make the computaional challenge of the UCB maximization step clearly in the paper and (2) further highlight the significance of the development of one-pass estimator:
>
> - $\underline{\mbox{One-pass estimator for GLB as the linear bandits setting}}$: We note that our estimator operates in a one-pass fashion, which not only performs updates **in $O(1)$ time complexity** per round but also **requires no storage of historical data**, retaining only the current estimator $\theta_t$ and Hessian $H_t$. In recent years, the design of one-pass estimators for GLB has attracted increasing attention [Zhang et al., 2016; Jun et al., 2017; Faury et al., 2022]. This direction is particularly compelling because, in the linear bandit setting, the least-squares estimator can be computed in a one-pass manner with O(1) time and memory per round. In contrast, the non-linearity of the link function in GLB introduces substantial computational challenges. *A natural question is whether it is possible to design GLB estimators that are as efficient as those in the linear case*, while still maintaining statistical efficiency. We provide an affirmative answer to this question.
> - $\underline{\mbox{Finite-arm case is common in practice}}$: From the practical side, our motivation for combining the one-pass estimator with the UCB strategy is that many real-world applications involve a finite number of arms. Examples include article recommendation tasks [1,2]. Moreover, many experimental evaluations of contextual bandits are also typically conducted in the finite-arm setting [3]. In such cases, training the estimator would become one of the major computational bottleneck.
> - $\underline{\mbox{Beyond LinUCB type method}}$: Finally, we would like to note that our one-pass estimator can serve as a versatile **plug-in component** applicable to a broad class of methods, not limited to LinUCB-type algorithms. For instance, following the same argument as  of Faury et al. (2022 Appendix D.2), the estimator can be applied to **Thompson Sampling**, where the arm selection step reduces to a convex optimization problem.
>
> [1] An Unbiased Offline Evaluation of Contextual Bandit Algorithms with
> Generalized Linear Models. JMLR 2012.
>
> [2] A contextual-bandit approach to personalized news article recommendation. WWW 2010.
>
> [3] A Contextual Bandit Bake-off. JMLR 2018.
>
> ---
>
> **Q2** : some of the claims made are not fully supported or slightly misleading.
>
> **A2:** We will clarify the statements as follows:
>
> > "not noting that OFU‑GLB [Lee] also handles unbounded GLMs"
>
> Thank you for the comments. we will clearly mention this in the next version.
>
> > claiming self‑concordant GLBs are “no more difficult” than linear bandits without accounting for the potentially huge constant term
>
> You are right, this expression is a bit sloppy as the second term contains the large constant $\kappa$. We will revise the sentence as: "...self-concordant GLBs **do not significantly exceed** the complexity of linear bandits, as our method achieves comparable regret and computational efficiency to OFUL for linear bandits **for sufficiently large $T$**"
>
> ---
>
> **Q3** : In particular, Proposition 2 is not a new result.
>
> **A3:** Thank you for letting us know. We believe the reviewer is referring to Lemma 2 in our paper. As noted by *Clerico et al., 2025* (in the paragraph following Proposition 2.1), this proposition can be traced back to *Ville (1939)* and is also known as the “no-hypercompression inequality.” We were not previously aware of the proposition’s historical depth and will make sure to clearly acknowledge this in the revised version.
>
> ---
>
> **Q4** : Could the method yield tighter, non‑ellipsoidal confidence sets by changing the OMD regularizer, instead of restricting to (looser) ellipsoids?
>
> **A4**: Yes. Based on the analysis in Section 4, it is indeed possible to construct a tighter confidence set of the form
>
> $\mathcal{C}\_t = \\left\\{\theta\in \mathbb{R}^d \mid \sum\_{s=1}^{t-1} \ell_s(\theta) \leq \sum\_{s=1}^{t-1} m_s(P_s) + \log \frac{1}{\delta} \\right\\},$
>
> where $P_s$ denotes a Gaussian distribution with mean $\theta_s$.
>
> The primary motivation for using ellipsoidal confidence sets **lies in computational efficiency** as they allow for a closed-form expression to compute $\mathrm{UCB}_t$ with a per-arm cost of $O(d^2)$. In contrast, constructing UCBs from non-ellipsoidal confidence sets requires solving a convex optimization problem for each arm with a computational cost of $O(t)$ per iteration.
>
> Moreover, ellipsoidal sets are more storage-efficient. They only require maintaining the mean and covariance, enabling one-pass updates. Non-ellipsoidal sets, on the other hand, typically require storing the full history of observations, with space complexity growing linearly in $T$.
>
> ---
>
> **Q5 :** substituting the log-likelihood loss for the mix-loss to derive confidence sets has appeared in a couple of very recent works
>
> **A5:** Thank you for bringing these concurrent papers to our attention. After a quick review, we found that the mix loss is indeed used in Section 3.3 of *Kirschner et al., 2025* and *Clerico et al., 2025* to derive confidence sets. The main distinction lies in the choice of comparator: their confidence sets are derived **based on the MLE**, while ours are built around **the OMD estimator**.
>
> This difference in objectives leads to distinct constructions of the distribution $P_s$ and subsequent analysis. Specifically, the prior works construct $P_s$ using exponential weights, whereas we take $P_s$ to be a Gaussian distribution. Naturally, the confidence sets in *Kirschner et al.* and *Clerico et al.* are tighter, but our ellipsoidal confidence sets based on the OMD framework offer better computational efficiency. We will properly highlight the similarities and differences with these concurrent works in the next revision.

---

> > ### Comment · Reviewer_7rDn · 2025-08-04
> >
> > Thank you for answering my questions. I’m happy to recommend acceptance, so I will raise my score to 5. See below for some more detailed comments.
> >
> > **Q1:**
> >
> > Thank you for your answer. The stated reasons have convinced that the development of (confidence sets for) a one-pass estimator is significant enough. As long as the cost of maximising the UCB is at least briefly mentioned somewhere, then I’m satisfied.
> >
> > **Q2:**
> >
> > I’m happy with the proposed changes.
> >
> > **Q3:**
> >
> > I’m happy with the proposed changes.
> >
> > **Q4:**
> >
> > Thank you confirming this. Given that a major focus of this work is computational efficiency, opting for quadratic regularisation/confidence sets makes sense.
> >
> > **Q5:**
> >
> > I think that the confidence sets in Clerico et al. (2025) would remain valid if the comparator was changed to something other than the MLE (the regret bound in their Proposition 3.1 holds for any comparator). Nevertheless, the implications of using the OMD-estimator as the comparator are not considered in Clerico et al. (2025), and I agree that the way that the mix-loss is used in this work is different.

---

> > > ### Author Response · Authors · 2025-08-05
> > >
> > > Thank you again for the expert review and for raising the score! Since the EWA method is used to construct the distribution $q_s$ in Clerico et al. (2025), the comparator can indeed be chosen beyond the MLE (though it seems a fixed $\bar{\theta}$ across all individual functions $\ell_s$ may still be required?). In our case, the comparison is made with the OMD estimator $\theta_s$, which varies across individual functions as the compared loss is $\ell_s(\theta_s)$. In the revision, we will include a discussion on the UCB calculation and make a more detailed comparison with the previous and concurrent work.

---

> > > > ### Comment · Reviewer_7rDn · 2025-08-05
> > > >
> > > > Yes, I think you're right that $\bar{\theta}$ needs to be fixed across all individual functions $\ell_s$. Good catch!

---

> > > > > ### Author Response · Authors · 2025-08-07
> > > > >
> > > > > Thank you! We will incorporate a more thorough discussion of the related work in the revision.

---

### Decision · Program_Chairs · 2025-09-17

**Decision:**

Accept (poster)

**Comment:**

The paper proposes a new, UCB-type generalized linear bandit algorithm that efficiently updates the estimate and confidence set of the unknown parameter via OMD. The authors leverage the self-concordance property of GLMs and introduce the mix-loss to decompose the inverse regret within the confidence radius, thereby establishing a tight confidence set and attaining nearly optimal regret. The proposed algorithm improves on the state of the art in at least one aspect, either by reducing computational complexity or by improving the regret bound.
Reviewers agree that the paper merits acceptance but suggest improvements in writing clarity and organization for the final version.